# PRIVATE ZEROTH-ORDER NONSMOOTH NONCONVEX OPTIMIZATION

**Qinzi Zhang, Hoang Tran & Ashok Cutkosky**
Department of Electrical and Computer Engineering
Boston University
Boston, MA, USA
{qinziz,tranhp,cutkosky}@bu.edu

## ABSTRACT

We introduce a new zeroth-order algorithm for private stochastic optimization on nonconvex and nonsmooth objectives. Given a dataset of size $M$, our algorithm ensures $(\alpha, \alpha\rho^2/2)$-Rényi differential privacy and finds a $(\delta, \epsilon)$-stationary point so long as $M = \tilde{\Omega}\left(\frac{d}{\delta\epsilon^3} + \frac{d^{3/2}}{\rho\delta\epsilon^2}\right)$. This matches the optimal complexity of its non-private zeroth-order analog. Notably, although the objective is not smooth, we have privacy "for free" whenever $\rho \geq \sqrt{d}\epsilon$.

## 1 INTRODUCTION

We study the stochastic optimization problem of the form

$$\min_{\boldsymbol{x} \in \mathbb{R}^d} F(\boldsymbol{x}) = \mathbb{E}_z[f(\boldsymbol{x}, z)],$$

where the stochastic function $f(\boldsymbol{x}, z)$ might be both non-convex and non-smooth with respect to $\boldsymbol{x}$. Our focus is on zeroth-order algorithms, often referred to as gradient-free algorithms. Unlike first-order algorithms that have access to the stochastic gradient $\nabla f(\boldsymbol{x}, z)$, zeroth-order algorithms can access only the function value $f(\boldsymbol{x}, z)$.

Non-convex stochastic optimization is fundamental to modern machine learning. For example, in deep learning, $\boldsymbol{x}$ is the weight of some neural network model, $z$ is a datapoint, and $f(\boldsymbol{x}, z)$ represents the loss of the model evaluated on $\boldsymbol{x}$ and $z$. Consequently, training the machine learning model equates to minimizing the non-convex objective $F(\boldsymbol{x})$. Given its significant role, there has been a marked increase in research focusing on non-convex optimization in recent years (Ghadimi & Lan, 2013; Arjevani et al., 2019; 2020; Carmon et al., 2019; Fang et al., 2018; Cutkosky & Orabona, 2019). Since finding a global minimum of non-convex $F(\boldsymbol{x})$ is intractable, many previous works assume that $F$ is smooth, and their goal is to find an $\epsilon$-stationary point $\boldsymbol{x}$ where $\|\nabla F(\boldsymbol{x})\| \leq \epsilon$. However, objectives are not always smooth, especially with the ever-increasing complexity of modern machine learning models. Making the problem even more difficult, recent research shows that finding an $\epsilon$-stationary point of a non-smooth objective might be intractable (Zhang et al., 2020; Kornowski & Shamir, 2022).

To address this issue, Zhang et al. (2020) introduces an alternative objective for the convergence analysis of non-smooth non-convex optimization. Specifically, they propose the notion of Goldstein stationary point (Goldstein, 1977): $\boldsymbol{x}$ is said to be an $(\delta, \epsilon)$-stationary point of $F$ if there exists a subset $S$ in the ball of radius $\delta$ centered at $\boldsymbol{x}$ such that $\|\mathbb{E}[\nabla F(\boldsymbol{y})]\| \leq \epsilon$ where $\boldsymbol{y}$ is uniformly distributed over $S$. Recently, there have been many works on first-order algorithms using this framework (Zhang et al., 2020; Tian et al., 2022; Cutkosky et al., 2023). Notably, Cutkosky et al. (2023) designs an "Online-to-non-convex Conversion" algorithm that finds a $(\delta, \epsilon)$-stationary point with $O(\delta^{-1}\epsilon^{-3})$ samples, which is proven to be the optimal rate. However, first-order algorithms have their own limitations. For many applications, computing first-order gradients can be computationally expensive or even impossible (Nelson, 2010; Mania et al., 2018). Hence, there are many recent results focusing on designing efficient zeroth-order algorithms for non-smooth non-convex optimization (Lin et al., 2022; Chen et al., 2023; Kornowski & Shamir, 2023). Recently, Kornowski & Shamir (2023) proposes a zeroth-order algorithm that requires only $O(d\delta^{-1}\epsilon^{-3})$ samples, which is

the current state-of-the-art rate for zeroth-order optimization on non-smooth non-convex objectives. The additional dimension dependence is an inevitable cost of zeroth-order algorithms even when the objective is smooth or convex (Duchi et al., 2015).

In this paper, we study zeroth-order stochastic optimization on non-convex and non-smooth objectives. Zeroth-order optimization is particularly important in cases where memory is constrained or the model is excessively large so that computing a backwards pass is forbiddingly expensive. In addition, we also aim to protect the privacy of user's data. For example, consider the practical problem of training a recommendation model or a chat response model. At each step the algorithm is given a set of different users who report the model's performance in the form of "upvotes" or "downvotes". It's important to make updates using such zeroth-order feedback while preserving the privacy of feedback from the individual users.

To this end, we require our algorithm to be *differentially private* (DP) (Dwork et al., 2006), which means with high probability, the output of our algorithm operating on any particular dataset is *almost* indistinguishable from the output when one datapoint in the dataset is perturbed.

The primary objective of private stochastic optimization is to minimize the dataset size required to solve the optimization problem while maintaining differential privacy guarantees. This area has seen extensive research efforts. While problems with convex objectives have been well-studied over the past decade (Chaudhuri et al., 2011; Jain et al., 2012; Kifer et al., 2012; Bassily et al., 2014; Talwar et al., 2014; Jain & Thakurta, 2014; Bassily et al., 2019; Feldman et al., 2020; Bassily et al., 2021b; Zhang et al., 2022), more recent efforts have focused on private non-convex optimization (Wang et al., 2017; 2019a; Tran & Cutkosky, 2022; Wang et al., 2019b; Zhou et al., 2020; Bassily et al., 2021a; Arora et al., 2023). Recently, Arora et al. (2023) designs an algorithm that, assuming the objective is smooth, finds an $\epsilon$-stationary point and satisfies $(\rho, \gamma)$-DP with $\tilde{O}(\epsilon^{-3} + \sqrt{d}\rho^{-1}\epsilon^{-2})$ data complexity. While private non-convex optimization has seen rapid advancements, certain challenges remain. Many previous studies hinge on the assumption of smooth objectives since their methods are essentially extensions of non-private non-convex optimization algorithms. In fact, this limitation emerges even in situations where the objective is convex, in which case smoothness provides a critical tool for reducing sensitivity. Since the results in non-smooth non-convex optimization are recent, its differentially private counterpart remains largely unexplored.

## 1.1 OUR CONTRIBUTIONS

The main contribution of this paper is the introduction of a zeroth-order algorithm for differentially private stochastic optimization on non-convex and non-smooth objectives. To our knowledge, this is the first work under this framework. Our algorithm satisfies $(\alpha, \alpha\rho^2/2)$-Rényi differential privacy (RDP) (Mironov, 2017) (which is approximately $(\rho, \gamma)$-DP) and finds a $(\delta, \epsilon)$-stationary point with $\tilde{O}(d\delta^{-1}\epsilon^{-3} + d^{3/2}\rho^{-1}\delta^{-1}\epsilon^{-2})$ data complexity. Notably, the non-private term $O(d\delta^{-1}\epsilon^{-3})$ matches the state-of-the-art complexity found in its non-private counterpart. Moreover, when $\rho \geq \sqrt{d}\epsilon$, the non-private term is dominating, suggesting that privacy can be attained without additional cost. This is consistent with the observations when the objective is smooth (Arora et al., 2023).

Our algorithm incorporates four essential components, each crucial for achieving the optimal rate. We leverage the non-private Online-to-non-convex Conversion (O2NC) framework by Cutkosky et al. (2023), which finds a $(\delta, \epsilon)$-stationary point of a non-smooth objective using a first-order oracle. We then build an approximate first-order oracle (i.e. a gradient estimator) with a zeroth-order oracle. Although this high-level strategy is a common technique, our approach distinguishes itself through its gradient oracle design. In contrast to prior non-private approaches that directly approximate the gradient with a standard zeroth-order estimator (Kornowski & Shamir, 2023), we introduce a variance-reduced gradient oracle. Our approach incorporates two zeroth-order estimators: one for the gradient and another for its difference between two points. Our gradient oracle also differs subtly from the standard zeroth-order estimator by sampling $d$ i.i.d. estimators for each data point, which is required to achieve the optimal dimension dependence. Both estimators exhibit reduced sensitivity. We reduce the privacy cost of our variance-reduced estimators by using the "tree mechanism" (Dwork et al., 2010; Chan et al., 2011), yielding a new state-of-the-art privacy guarantee.

Omitting any component from our algorithm significantly impacts the results. Neither utilizing the O2NC technique nor sampling $d$ i.i.d. estimators results in an additional dimension dependence of

$\sqrt{d}$. Moreover, the combination of the variance-reduced oracle and the tree mechanism is crucial for optimal privacy. A naive approach might directly make O2NC private adding noise to classical zeroth-order gradient estimators, but this yields a sub-optimal rate with an extra factor of $\epsilon^{-1}$. More details about the intuitions and challenges of our algorithm can be found in Section 4.1.

## 2 PRELIMINARY

**Notation**   We use bold font $x$ to denote a vector $x \in \mathbb{R}^d$, and denote the Euclidean norm of $x$ by $\|x\|$. We denote the unit ball and unit sphere in $\mathbb{R}^d$ by $\mathbb{B}$ and $\mathbb{S}$, and denote the uniform distribution on $\mathbb{B}$ and $\mathbb{S}$ by $\mathcal{U}_\mathbb{B}$ and $\mathcal{U}_\mathbb{S}$ respectively. We denote an open ball in $\mathbb{R}^d$ centered at $x$ of radius $r$ by $B(x, r) = \{y : \|x - y\| < r\}$. For $n \in \mathbb{N}$, we denote the set $\{1, \ldots, n\}$ by $[n]$, and we denote a sequence $a_1, \ldots, a_n$ by $a_{1:n}$. We use the standard big-O notation, and use $\tilde{O}$ to hide additional logarithmic factors. We interchangeably use $f(x) \lesssim g(x)$ to denote $f(x) = \tilde{O}(g(x))$.

**Non-smooth Optimization**   A function $h : \mathbb{R}^d \to \mathbb{R}$ is $L$-Lipschitz if $|h(x) - h(y)| \leq L\|x - y\|$ for all $x, y \in \mathbb{R}^d$; $h$ is $H$-smooth if it is differentiable and $\|\nabla h(x) - \nabla h(y)\| \leq H\|x - y\|$ for all $x, y \in \mathbb{R}^d$. A point $x^* \in \mathbb{R}^d$ is an $\epsilon$-stationary point of $h$ if $\|\nabla h(x^*)\| \leq \epsilon$.

**Definition 2.1.** Let $\delta > 0$ and $h : \mathbb{R}^d \to \mathbb{R}$ be differentiable. The Goldstein $\delta$-subdifferential of $h$ at $x$ is $\partial_\delta h(x) = \mathrm{conv}(\cup_{y \in B(x, \delta)} \nabla h(y))$. We define $\|\nabla h(x)\|_\delta = \inf\{\|g\| : g \in \partial_\delta h(x)\}$. Then $x^*$ is said to be a Goldstein $(\delta, \epsilon)$-stationary point of $h$ if $\|\nabla h(x^*)\|_\delta \leq \epsilon$. Note that

$$\|\nabla h(x)\|_\delta \leq \inf_{S \subset B(x, \delta)} \left\| \frac{1}{|S|} \sum_{y \in S} \nabla h(y) \right\|.$$

**Differential Privacy**   A stochastic optimization algorithm can be considered as a randomized algorithm that takes a dataset $Z$ (a collection of data points $z_1, \ldots, z_M$) and outputs an output $\overline{w}$. Throughout this paper, we denote $\mathcal{Z}$ as the set of all possible datasets of size $M$. Two datasets $Z, Z' \in \mathcal{Z}$ are neighboring if they differ only in one data point. A randomized algorithm $\mathcal{A} : \mathcal{Z} \to \mathcal{R}$ is said to be $(\epsilon, \delta)$-differentially private ($(\epsilon, \delta)$-DP) for $\epsilon, \delta > 0$ if for all neighboring datasets $Z, Z'$ and measurable $E \subseteq \mathcal{R}$, we have $\mathbb{P}\{\mathcal{A}(Z) \in E\} \leq e^\epsilon \mathbb{P}\{\mathcal{A}(Z') \in E\} + \delta$ (Dwork et al., 2006).

Another common privacy measure is Rényi differential privacy (RDP). Algorithm $\mathcal{A}$ is $(\alpha, \rho)$-RDP for $\alpha > 1, \rho > 0$ if for all neighboring $Z, Z'$, $D_\alpha(\mathcal{A}(Z)\|\mathcal{A}(Z')) \leq \rho$ (Mironov, 2017), where $D_\alpha(\mu\|\nu)$ is the Rényi divergence of distributions $\mu, \nu$. RDP can be converted to $(\epsilon, \delta)$-DP as follows: if an algorithm is $(\alpha, \alpha\rho^2/2)$-RDP for all $\alpha > 1$, then it is also $(2\rho \ln(1/\delta)^{1/2}, \delta)$-DP for all $\delta \geq \exp(-\rho^2)$ (Mironov, 2017, Proposition 3). Therefore, we use $(\alpha, \alpha\rho^2/2)$-RDP as a measure of differential privacy in our paper.

If $\|\mathcal{A}(Z) - \mathcal{A}(Z')\| \leq s$ for any neighboring $Z, Z'$, we say the sensitivity of $\mathcal{A}$ is bounded by $s$. It is well known that in this case, adding a Gaussian noise $\mathcal{N}(0, \sigma^2 I)$ to the output of $\mathcal{A}$, where $\sigma = s/\rho$, ensures that $\mathcal{A}$ is $(\alpha, \alpha\rho^2/2)$-RDP (Mironov, 2017).

We also make use of the "tree mechanism" Dwork et al. (2010); Chan et al. (2011), which is a technique that allows for private release of *running sums* of potentially sensitive data (Algorithm 5).

**Online Learning**   Our algorithm builds on the Online-to-non-convex algorithm by Cutkosky et al. (2023), so we briefly introduce the setting of online convex optimization (OCO) (Cesa-Bianchi & Lugosi, 2006; Hazan, 2019; Orabona, 2019). An OCO algorithm proceeds in rounds. In each round $t$, it outputs $x_t$, receives a convex loss function $\ell_t(x)$, and suffers loss $\ell_t(x_t)$. It is common to use a linear loss $\ell_t(x) = \langle v_t, x \rangle$. An OCO algorithm has domain bounded by $D$ if $\|x_t\| \leq D$ for all $t$.

The goal of online learning is to minimize the *static regret* defined as

$$\mathrm{Reg}_T(u) = \sum_{t=1}^T \ell_t(x_t) - \ell_t(u) = \sum_{t=1}^T \langle v_t, x_t - u \rangle,$$

where $u$ is some comparator vector (often, $u = \arg\min \sum_{t=1}^T \ell_t(x)$). There are many OCO algorithms that achieve the minimax optimal $O(\sqrt{T})$ regret. For example, projected Online Subgradient Descent (OSD) (Zinkevich, 2003) with domain bounded by $D$ satisfies

$$\mathbb{E}[\mathrm{Reg}_T(x)] \leq D\sqrt{\sum_{t=1}^T \mathbb{E}\|v_t\|^2}.$$

**Uniform Smoothing**  Randomized smoothing is a well known technique that converts a possibly non-smooth function to a smooth approximation (Flaxman et al., 2005; Duchi et al., 2012; Lin et al., 2022). Given a Lipschitz function $h : \mathbb{R}^d \to \mathbb{R}$ and $\delta > 0$, we define the *uniform smoothing* of $h$ as $\hat{h}_\delta(\boldsymbol{x}) = \mathbb{E}_{\boldsymbol{v} \sim \mathcal{U}_\mathbb{B}}[h(\boldsymbol{x} + \delta \boldsymbol{v})]$. In later sections, we denote $\hat{F}_\delta(\boldsymbol{x})$ and $\hat{f}_\delta(\boldsymbol{x}, z)$ as the uniform smoothing of the objective $F(\boldsymbol{x})$ and $f(\boldsymbol{x}, z)$ respectively. As shown in prior works (see (Yousefian et al., 2012) and Duchi et al. (2012, Section E)), uniform smoothing is a smooth approximation whose gradient can be estimated by finite differentiation. We rephrase the key properties in the following lemma, whose proof is presented in the appendix for completeness.

**Lemma 2.2.** *Suppose $h : \mathbb{R}^d \to \mathbb{R}$ is L-Lipschitz. Then (i) $\hat{h}_\delta$ is L-Lipschitz; (ii) $\|\hat{h}_\delta(\boldsymbol{x}) - h(\boldsymbol{x})\| \leq L\delta$; (iii) $\hat{h}_\delta$ is differentiable and $\frac{\sqrt{d}L}{\delta}$-smooth; (iv)*

$$\nabla \hat{h}_\delta(\boldsymbol{x}) = \mathbb{E}_{\boldsymbol{u} \sim \mathcal{U}_\mathbb{S}}[\tfrac{d}{\delta} h(\boldsymbol{x} + \delta \boldsymbol{u}) \boldsymbol{u}] = \mathbb{E}_{\boldsymbol{u} \sim \mathcal{U}_\mathbb{S}}[\tfrac{d}{2\delta}(h(\boldsymbol{x} + \delta \boldsymbol{u}) - h(\boldsymbol{x} - \delta \boldsymbol{u}))\boldsymbol{u}];$$

As a corollary, finding an $(\delta, \epsilon)$-stationary point of $\hat{F}_\delta$ is sufficient to find an $(2\delta, \epsilon)$-stationary point of $F$. The formal statement is as follows (Kornowski & Shamir (2023, Lemma 4); proof is presented in Appendix A for completeness).

**Corollary 2.3.** *Suppose $F : \mathbb{R}^d \to \mathbb{R}$ is L-Lipschitz. Then for any $\epsilon, \delta > 0$, $\|\nabla \hat{F}_\delta(\boldsymbol{x})\|_\delta \leq \epsilon$ implies that $\|\nabla F(\boldsymbol{x})\|_{2\delta} \leq \epsilon$.*

## 2.1  Analysis Organization

As a brief overview, our algorithm leverages O2NC to privately optimize the uniform smoothing of the objective. Note that O2NC does not require any smoothness itself - we employ uniform smoothing because it allows us to construct a low-sensitivity gradient oracle from a zeroth-order oracle. The smoothness property is tangential. We construct a variance-reduced gradient oracle for O2NC and incorporate the tree mechanism for enhanced privacy. This oracle is based on two zeroth-order estimators: one for the stochastic gradient of the uniform smoothing and another for the gradient difference.

This paper is organized in the following order. In Section 3, we introduce the zeroth-order estimators. In Section 4.1, we present the O2NC algorithm combined with the private variance-reduced gradient oracle. In Section 4.2, we demonstrate the improved privacy guarantee provided by the tree mechanism. We conclude with the convergence analysis in Section 4.3.

## 3  Zeroth-Order Estimators

Let $F(\boldsymbol{x}) = \mathbb{E}_z[f(\boldsymbol{x}, z)]$ and denote $\hat{F}_\delta$ and $\hat{f}_\delta$ as the uniform smoothing of $F$ and $f$ respectively. Using the randomized smoothing technique from Lemma 2.2, we can construct unbiased zeroth-order estimators for both $\nabla \hat{F}_\delta(\boldsymbol{x})$ and $\nabla \hat{F}_\delta(\boldsymbol{x}) - \nabla \hat{F}_\delta(\boldsymbol{y})$. The key properties of these estimators GRAD and DIFF, as defined in Algorithm 1 and 2, are summarized in the following results.

**Lemma 3.1.** *If $f(\boldsymbol{x}, z)$ is differentiable and L-Lipschitz in $\boldsymbol{x}$, then for any $\delta > 0, \boldsymbol{x} \in \mathbb{R}^d$ and neighboring data batches $z_{1:b}, z'_{1:b}$ of size $b$,*

$$\mathbb{E}[\text{GRAD}_{f,\delta}(\boldsymbol{x}, z_{1:b})] = \nabla \hat{F}_\delta(\boldsymbol{x}), \tag{unbiased}$$

$$\mathbb{E}\|\text{GRAD}_{f,\delta}(\boldsymbol{x}, z_{1:b}) - \nabla \hat{F}_\delta(\boldsymbol{x})\|^2 \leq \tfrac{16dL^2}{b}, \tag{variance}$$

$$\|\text{GRAD}_{f,\delta}(\boldsymbol{x}, z_{1:b}) - \text{GRAD}_{f,\delta}(\boldsymbol{x}, z'_{1:b})\| \leq \tfrac{2dL}{b}. \tag{sensitivity}$$

**Lemma 3.2.** *If $f(\boldsymbol{x}, z)$ is differentiable and L-Lipschitz in $\boldsymbol{x}$, then for any $\delta > 0$, $\boldsymbol{x}, \boldsymbol{y} \in \mathbb{R}^d$ and neighboring data batches $z_{1:b}, z'_{1:b}$ of size $b$,*

$$\mathbb{E}[\text{DIFF}_{f,\delta}(\boldsymbol{x}, \boldsymbol{y}, z_{1:b})] = \nabla \hat{F}_\delta(\boldsymbol{x}) - \nabla \hat{F}_\delta(\boldsymbol{y}), \tag{unbiased}$$

$$\mathbb{E}\|\text{DIFF}_{f,\delta}(\boldsymbol{x}, \boldsymbol{y}, z_{1:b}) - [\nabla \hat{F}_\delta(\boldsymbol{x}) - \nabla \hat{F}_\delta(\boldsymbol{y})]\|^2 \leq \tfrac{16dL^2}{b\delta^2}\|\boldsymbol{x} - \boldsymbol{y}\|^2, \tag{variance}$$

$$\|\text{DIFF}_f(\boldsymbol{x}, \boldsymbol{y}, z_{1:b}) - \text{DIFF}_f(\boldsymbol{x}, \boldsymbol{y}, z'_{1:b})\| \leq \tfrac{2dL}{b\delta}\|\boldsymbol{x} - \boldsymbol{y}\|. \tag{sensitivity}$$

---

**Algorithm 1** Zeroth-order gradient oracle $\text{GRAD}_{f,\delta}(\boldsymbol{x}, z_{1:b})$

---

**Input:** loss function $f$, constant $\delta > 0$, parameter $\boldsymbol{x}$, i.i.d. data batch $z_{1:b}$
1: Sample $\boldsymbol{u}_{11}, \ldots, \boldsymbol{u}_{bd} \sim \text{Uniform}(\mathbb{S})$ i.i.d.
2: Return $\text{GRAD}_{f,\delta}(\boldsymbol{x}, z_{1:b}) = \frac{1}{b} \sum_{i=1}^{b} \frac{1}{d} \sum_{j=1}^{d} \frac{d}{2\delta}(f(\boldsymbol{x} + \delta\boldsymbol{u}_{ij}, z_i) - f(\boldsymbol{x} - \delta\boldsymbol{u}_{ij}, z_i))\boldsymbol{u}_{ij}$.

---

**Algorithm 2** Zeroth-order gradient difference oracle $\text{DIFF}_{f,\delta}(\boldsymbol{x}, \boldsymbol{y}, z_{1:b})$

---

**Input:** loss function $f$, constant $\delta > 0$, parameter $\boldsymbol{x}, \boldsymbol{y}$, i.i.d. data batch $z_{1:b}$
1: Sample $\boldsymbol{u}_{11}, \ldots, \boldsymbol{u}_{bd} \sim \text{Uniform}(\mathbb{S})$ i.i.d.
2: Return $\text{DIFF}_{f,\delta}(\boldsymbol{x}, \boldsymbol{y}, z_{1:b}) = \frac{1}{b} \sum_{i=1}^{b} \frac{1}{d} \sum_{j=1}^{d} \frac{d}{\delta}(f(\boldsymbol{x} + \delta\boldsymbol{u}_{ij}, z_i) - f(\boldsymbol{y} + \delta\boldsymbol{u}_{ij}, z_i))\boldsymbol{u}_{ij}$.

---

Proofs for Lemmas 3.1 and 3.2 are both stemming from the uniform smoothing properties detailed in Lemma 2.2. We defer the proofs to Appendix B. Notably, these lemmas indicate that both the gradient and gradient difference estimators are unbiased, with low variance and sensitivity. Specifically, in our final algorithm, we will be able to control the distance between parameters $\boldsymbol{x}, \boldsymbol{y}$ in two successive iterations by $\delta/T$. As a result, the variance in Lemma 3.2 is bounded by $O(dL^2/bT^2)$, and its sensitivity is bounded by $O(dL/bT)$. Using these estimators, we construct zeroth-order gradient oracles by exploiting the decomposition $\nabla \hat{F}_\delta(\boldsymbol{x}_t) = \nabla \hat{F}_\delta(\boldsymbol{x}_1) + \sum_{i=1}^{t} \nabla \hat{F}_\delta(\boldsymbol{x}_i) - \nabla \hat{F}_\delta(\boldsymbol{x}_{i-1}) \approx \text{GRAD}_{f,\delta}(\boldsymbol{x}_1, z) + \sum_{i=1}^{t} \text{DIFF}_{f,\delta}(\boldsymbol{x}_i, \boldsymbol{x}_{i-1}, z)$. Our constructed oracles have low variance and low sensitivity, thus allowing us to incorporate the tree mechanism to privately aggregate the sum with less noise. A more detailed exploration of this idea is presented in the subsequent sections, especially in Corollary 4.4 and Lemma 4.8.

In addition, we would like to discuss the rationale behind sampling $d$ i.i.d. uniform vectors $\boldsymbol{u}_{ij}$ for each data point $z_i$. For the gradient estimator GRAD, it is feasible to approximate $\nabla \hat{f}_\delta(\boldsymbol{x}, z_i)$ with a single two-point estimator, namely $\boldsymbol{g}_i = \frac{d}{2\delta}(f(\boldsymbol{x} + \delta\boldsymbol{u}, z_i) - f(\boldsymbol{x} - \delta\boldsymbol{u}, z_i))\boldsymbol{u}$, as it is proved that $\mathbb{E}\|\boldsymbol{g}_i\|^2 = O(dL^2)$ (Shamir, 2017, Lemma 10). The central argument hinges on a concentration inequality for Lipschitz functions, which states that if $h$ is $L$-Lipschitz on $\boldsymbol{u}$ and $\boldsymbol{u} \sim \mathcal{U}_\mathbb{S}$, then $\mathbb{P}\{|h(\boldsymbol{u}) - \mathbb{E}h(\boldsymbol{u})| \geq t\} \leq 2\exp(-\frac{dt^2}{2L^2})$, as per (Wainwright, 2019) Proposition 3.11. In the context of their proof, $h(\boldsymbol{u}) = f(\boldsymbol{x} + \delta\boldsymbol{u})$ which is $L\delta$-Lipschitz.

However, this argument isn't applicable to the gradient difference estimator DIFF. In order to apply the concentration inequality, we need to bound the Lipschitz constant of $h(\boldsymbol{u}) = f(\boldsymbol{x} + \delta\boldsymbol{u}) - f(\boldsymbol{y} + \delta\boldsymbol{u})$. Although this function is indeed $2L\delta$-Lipschitz, using this would bound the variance at $O(dL^2)$, which doesn't match our target of $O(dL^2/T^2)$. On the other hand, directly using the Lipschitzness of $f$ bounds $\mathbb{E}\|\frac{d}{\delta}(f(\boldsymbol{x}+\delta\boldsymbol{u}, z_i)-f(\boldsymbol{y}+\delta\boldsymbol{u}, z_i))\boldsymbol{u}\|^2 \leq (\frac{dL}{\delta}\|\boldsymbol{x}-\boldsymbol{y}\|)^2 = O(d^2L^2/T^2)$. Consequently, we need to sample $d$ i.i.d. uniform vectors to reduce the dimension dependence of the variance by a factor of $d$ in order to match the optimal rate. While a more efficient analysis that avoids sampling $d$ uniform vectors might exist, we are currently unaware of it.

## 4 OUR ALGORITHM AND RESULTS

### 4.1 PRIVATE ONLINE-TO-NON-CONVEX CONVERSION

Our algorithm builds on the Online-to-Nonconvex Conversion (O2NC) by (Cutkosky et al., 2023), which is a general framework that converts any OCO algorithm with $O(\sqrt{T})$ regret into a nonsmooth nonconvex optimization algorithm that finds a $(\delta, \epsilon)$-stationary point in $O(\delta^{-1}\epsilon^{-3})$ iterations. The pseudo-code of this framework is presented in Algorithm 3. Specifically, our algorithm chooses projected Online Subgradient Descent (OSD) as the OCO algorithm, which updates $\Delta_t^k$ in line 3 of Algorithm 3 following the explicit update rule: $\Delta_1^k \leftarrow 0$ and $\Delta_t^k \leftarrow \Pi_D(\Delta_{t-1}^k - \eta\tilde{\boldsymbol{g}}_{t-1}^k)$ for $t \geq 2$. Here $\Pi_D(\boldsymbol{x}) = \arg\min_{\|\boldsymbol{y}\| \leq D} \|\boldsymbol{x} - \boldsymbol{y}\|$ denotes the projection operator and $\eta$ is the stepsize tuned by OSD. Regarding notations in Algorithm 3, there are two round indices $t$ and $k$, and we use the subscript $t$ and superscript $k$, namely $\boldsymbol{x}_t^k$, to denote a variable $\boldsymbol{x}$ in outer loop $k$ and inner loop $t$.

The selection of O2NC as our base non-private algorithm is deliberate and influenced by the literature of non-private zeroth-order algorithms in nonsmooth nonconvex optimization. Intuitively,

---

**Algorithm 3** Online-to-Nonconvex Conversion

---

**Input:** OCO algorithm $\mathcal{A}$ with domain $D$, gradient oracle $\mathcal{O}$, initial state $\boldsymbol{x}_1$.
1: **for** $k = 1, 2, \ldots, K$ **do**
2:     **for** $t = 1, 2, \ldots, T$ **do**
3:         Receive $\Delta_t^k$ from $\mathcal{A}$. (For example, $\Delta_1^k \leftarrow 0$ and $\Delta_t^k \leftarrow \Pi_D(\Delta_{t-1}^k - \eta \tilde{\boldsymbol{g}}_{t-1}^k), t \geq 2$.)
4:         Update $\boldsymbol{x}_{t+1}^k \leftarrow \boldsymbol{x}_t^k + \Delta_t^k$ and $\boldsymbol{w}_t^k \leftarrow \boldsymbol{x}_t^k + s_t^k \Delta_t^k$, where $s_t^k \sim \text{Uniform}([0,1])$.
5:         Query gradient estimator $\tilde{\boldsymbol{g}}_t^k$ from $\mathcal{O}$.
6:         Send $\ell_t^k(\cdot) = \langle \tilde{\boldsymbol{g}}_t^k, \cdot \rangle$ to $\mathcal{A}$ and $\mathcal{A}$ updates $\Delta_{t+1}^k$.
7:     **end for**
8:     Compute $\overline{\boldsymbol{w}}^k \leftarrow \frac{1}{T} \sum_{t=1}^{T} \boldsymbol{w}_t^k$.
9:     Reset $\boldsymbol{x}_1^{k+1} \leftarrow \boldsymbol{x}_{T+1}^k$ and restart $\mathcal{A}$.
10: **end for**
11: Output $\overline{\boldsymbol{w}} \sim \text{Uniform}(\{\overline{\boldsymbol{w}}^1, \ldots, \overline{\boldsymbol{w}}^K\})$.

---

in the realm of privacy, a variance-reduction algorithm is more appealing due to its inherently low sensitivity. However, a straightforward application of variance-reduced SGD paired with zeroth-order estimators only achieves a sub-optimal dimension dependence of $O(d^{3/2}\delta^{-1}\epsilon^{-3})$, as shown in (Chen et al., 2023). A recent advancement by Kornowski & Shamir (2023) has improved the rate to $O(d\delta^{-1}\epsilon^{-3})$. Central to their achievement is the observation that finding a $(2\delta, \epsilon)$-stationary point of a nonsmooth objective $F$ is equivalent to finding a $(\delta, \epsilon)$-stationary point of its uniform smoothing $\hat{F}_\delta$. Previous zeroth-order algorithms aim to find an $\epsilon$-stationary point of $\hat{F}_\delta$ via smooth optimization algorithms such as variance-reduced SGD, resulting in an additional smoothness cost of order $O(\sqrt{d})$. In contrast, the algorithm proposed by Kornowski & Shamir (2023) leverages O2NC to find a $(\delta, \epsilon)$-stationary point, achieving a linear dimension dependence.

Informed by the insights of (Kornowski & Shamir, 2023), we adopt O2NC as our foundational non-private algorithm. Specifically, we adapt Algorithm 3 for privacy by substituting the gradient oracle $\mathcal{O}$ in line 5 with a more carefully designed private oracle. A main challenge in this design is to minimize *sensitivity*, which is required to ensure privacy. In the algorithm proposed by Kornowski & Shamir (2023), the gradient oracle is simply the zeroth-order gradient estimator, $\tilde{\boldsymbol{g}}_t^k = \frac{d}{2\delta}(f(\boldsymbol{w}_t^k + \delta\boldsymbol{u}, z_t^k) - f(\boldsymbol{w}_t^k + \delta\boldsymbol{u}, z_t^k))\boldsymbol{u}$, whose sensitivity is $O(dL)$. A straightforward method to ensure privacy for this algorithm is by directly adding Gaussian noise with variance $O(d^2 L^2/\rho^2)$. While this might seem intuitive, it leads to significantly worse data complexity. Specifically, the resulting rate is $O(d^{3/2}\rho^{-1}\delta^{-1}\epsilon^{-3})$, which is worse by a factor of $\epsilon^{-1}$ when compared to our proposed algorithm - and importantly *does not* admit any value of $\rho$ for which we obtain the non-private convergence rate. For interested readers, we include the detailed analysis of this naive approach in Appendix E.

In contrast, we designed a more refined private gradient oracle in Algorithm 4. According to Lemma 3.1 and 3.2, the sensitivity of $\boldsymbol{g}_1^k$ is $O(dL/B_1)$, while that of $\boldsymbol{d}_t^k$ is $O(dL\|\boldsymbol{w}_t^k - \boldsymbol{w}_{t-1}^k\|/B_2\delta)$. Setting $B_1 = T, B_2 = 1$, and using Remark 4.1 with $D = \delta/T$, both $\boldsymbol{g}_1^l$ and $\boldsymbol{d}_t^k$ have sensitivity $O(dL/T)$. With these settings, we can apply the tree mechanism, a useful technique to release sums of private algorithms, ensuring that the variance-reduced gradient $\tilde{\boldsymbol{g}}_t^k$ achieves $(\alpha, \alpha\rho^2/2)$-RDP. Specifically, the tree mechanism adds noise of order $\tilde{O}(d^2 L^2/\rho^2 T^2)$ to each $\tilde{\boldsymbol{g}}_t^k$. Compared to the aforementioned naive approach, our refined approach reduces noise by a factor of $T^2$, thus resulting in the desired data complexity.

*Remark* 4.1. Note that $\boldsymbol{w}_{t+1}^k = \boldsymbol{x}_{t+1}^k + s_{t+1}^k \Delta_{t+1}^k = \boldsymbol{w}_t^k + (1 - s_t^k)\Delta_t^k + s_{t+1}^k \Delta_{t+1}^k$. Therefore, if the OCO algorithm has domain bounded by $D$ (i.e., $\|\Delta_t^k\| \leq D$ for all $t, k$), then $\|\boldsymbol{w}_{t+1}^k - \boldsymbol{w}_t^k\| \leq 2D$. In general, for all $t, t' \in [T]$, $\|\boldsymbol{w}_t^k - \boldsymbol{w}_{t'}^k\| \leq 2|t - t'|D$.

## 4.2 PRIVACY GUARANTEE

To ensure the privacy of the Online-to-Nonconvex conversion, we use a variant of the tree mechanism, as defined in Algorithm 5, to privately aggregate the sum of $\boldsymbol{g}_1^k$ and $\boldsymbol{d}_t^k$. The privacy guarantee of the tree mechanism is presented in Theorem 4.3, and the proof is presented in Append C.1. Intuitively, the tree mechanism says if each of the algorithms $\mathcal{M}_1, \ldots, \mathcal{M}_n$ requires $\sigma^2$ noise for privacy, then the sum $\sum_{i=1}^{n} \mathcal{M}_i$ only needs $O(\ln(n)\sigma^2)$ noise for the same level of privacy. With Theorem

---

**Algorithm 4** Private variance-reduced gradient oracle $\mathcal{O}$

---

  **Input:** dataset $\mathcal{Z}$, constants $B_1, B_2$
  **Initialize:** Partition $\mathcal{Z}$ into $KT$ subsets: $Z_1^k$ of size $B_1$ for $k \in [K]$ and $Z_t^k$ of size $B_2$ for $t = [2, K], k \in [K]$. The total data size is $|\mathcal{Z}| = M = K(B_1 + B_2(T-1))$.
1: Upon receiving round index $k, t$ and parameters $\boldsymbol{w}_t^k$:
2: **if** $t = 1$ **then**
3:   Query $\boldsymbol{g}_1^k \leftarrow \text{GRAD}_{f,\delta}(\boldsymbol{w}_1^k, Z_1^k)$.
4: **else**
5:   Query $\boldsymbol{d}_t^k \leftarrow \text{DIFF}_{f,\delta}(\boldsymbol{w}_t^k, \boldsymbol{w}_{t-1}^k, Z_t^k)$ and compute $\boldsymbol{g}_t^k \leftarrow \boldsymbol{g}_{t-1}^k + \boldsymbol{d}_t^k$.
6: **end if**
7: Return $\tilde{\boldsymbol{g}}_t^k \leftarrow \tilde{\boldsymbol{g}}_t^k + \text{TREE}(t)$.

---

**Algorithm 5** Tree Mechanism

---

  **Input:** Index $t \in [T]$, global variables $\sigma_1, \ldots, \sigma_T$.
1: If $t = 1$, set NOISE $\leftarrow \{\}$.
2: Sample $\xi_t \sim N(0, \sigma_t^2 I)$ and store $\xi_t$ in NOISE.
3: Return $\sum_{(\cdot, i) \in \text{NODE}(t)} \xi_i$.

4: **function** NODE(t)
5:   **Initialize:** Set $k \leftarrow 0$ and $S \leftarrow \{\}$.
6:   **for** $i = 0, \ldots, \lceil \log_2(t) \rceil$ while $k < t$ **do**
7:     Set $k' \leftarrow k + 2^{\lceil \log_2(t) \rceil - i}$. If $k' \leq t$, store $S \leftarrow S \cup \{(k+1), k'\}$ and update $k \leftarrow k'$.
8:   **end for**
9:   Return $S$.
10: **end function**

---

4.3, we determine the minimal noise required to ensure privacy for Algorithm 3 in Corollary 4.4, and we evaluate the total noise added by the tree mechanism in Corollary 4.5.

*Remark* 4.2. To better understand the NODE function in Algorithm 5, consider a binary tree of depth $\lceil \log_2(t) \rceil$. NODE(t) returns the largest node $n_i$ in each layer $i$, if exists, such that $n_j < n_i \leq t$ starting from the top level. As an example, NODE(7) $= \{(1,4), (5,6), (7,7)\}$ and NODE(8) $= \{(1,8)\}$. In particular, note that $|\text{NODE}(t)| \leq \lceil \log_2(t) \rceil \leq 2 \ln(t)$.

Next, we present the main privacy theorem for the tree mechanism as presented in Algorithm 5.

**Theorem 4.3.** *Let $\mathcal{X}$ be state space and $\mathcal{Z}^{(1)}, \ldots, \mathcal{Z}^{(n)}$ be dataset spaces, and denote $\mathcal{Z}^{(1:i)} = \mathcal{Z}^{(1)} \times \cdots \times \mathcal{Z}^{(i)}$. Let $\mathcal{M}^{(i)} : \mathcal{X}^{i-1} \times \mathcal{Z}^{(i)} \to \mathcal{X}$ be a sequence of algorithms for $i \in [n]$, and let $\mathcal{A} : \mathcal{Z}^{(1:n)} \to \mathcal{X}^n$ be the algorithm that, given a dataset $Z_{1:n} \in \mathcal{Z}^{(1:n)}$, sequentially computes $\boldsymbol{x}_i = \sum_{j=1}^i \mathcal{M}^{(j)}(\boldsymbol{x}_{1:j-1}, Z_i) + \text{TREE}(i)$ for $i \in [n]$ and then outputs $\boldsymbol{x}_{1:n}$. Suppose for all $i \in [n]$ and neighboring $Z_{1:n}, Z_{1:n}' \in \mathcal{Z}^{(1:n)}$, $\|\mathcal{M}^{(i)}(\boldsymbol{x}_{1:i-1}, Z_i) - \mathcal{M}^{(i)}(\boldsymbol{x}_{1:i-1}, Z_i')\| \leq s_i$ for all auxiliary inputs $\boldsymbol{x}_{1:i-1} \in \mathcal{X}^{i-1}$. Then for all $\alpha > 1$, $\mathcal{A}$ is $(\alpha, \alpha\rho^2/2)$-RDP where*

$$\rho \leq \sqrt{2 \ln n} \cdot \max_{b \in [n], i \leq b} \frac{s_i}{\sigma_b}.$$

In our case, $\mathcal{X}$ is the collection of weights $\boldsymbol{x}$ for $f(\boldsymbol{x}, z)$, $\mathcal{Z}^{(1)}$ is the collection of all possible batches $Z_1^k$ of size $B_1$ defined in Algorithm 3 and $\mathcal{Z}^{(t)}$ for $t \geq 2$ is the collection of all possible batches $Z_t^k$ of size $B_2$. Moreover, $\mathcal{M}^{(1)}$ corresponds to GRAD that computes $\boldsymbol{g}_1^k$ and $\mathcal{M}^{(t)}$ corresponds to DIFF that computes $\boldsymbol{d}_t^k$. Upon substituting these specific definition into Theorem 4.3, we have the following privacy guarantees.

**Corollary 4.4.** *Suppose $f(\boldsymbol{x}, z)$ is differentiable and $L$-Lipschitz in $\boldsymbol{x}$ and the domain of $\mathcal{A}$ is bounded by $D = \delta/T$. Let $B_1, B_2$ satisfies $B_1 \geq TB_2/2$. Then for any $\alpha > 1, \rho > 0$, Algorithm 3 is $(\alpha, \alpha\rho^2/2)$-RDP if we set the noises $\sigma_{1:T}$ in the tree mechanism as*

$$\sigma_t = \sigma := \frac{\sqrt{2 \ln T} 4dL}{B_2 T \rho}.$$

**Corollary 4.5.** *Following the assumptions and definitions in Corollary 4.4, for all $t \in [T]$,*

$$\mathbb{E}\|\text{TREE}(t)\|^2 \le 2\ln(t)d\sigma^2 \le \left(\frac{8\ln(T)d^{3/2}L}{B_2 T \rho}\right)^2.$$

## 4.3 CONVERGENCE ANALYSIS

To start the convergence analysis, we revisit the convergence result of the non-private O2NC algorithm as presented in Lemma 4.6 whose proof is included in Appendix D for completeness. Building upon the observation in Corollary 2.3, our objective shifts to finding a $(\delta, \epsilon)$-stationary point of the uniform smoothing $\hat{F}_\delta$, as opposed to directly optimizing $F$. This approach leads to the formulation of Corollary 4.7. Missing proofs in this section are deferred to Appendix D.

**Lemma 4.6.** *For any function $F : \mathbb{R}^d \to \mathbb{R}$ that is differentiable and $F(\boldsymbol{x}_1) - \inf_{\boldsymbol{x}} F(\boldsymbol{x}) \le F^*$, if the domain of the OCO algorithm $\mathcal{A}$ is bounded by $D = \delta/T$, then*

$$\mathbb{E}\|\nabla F(\overline{\boldsymbol{w}})\|_\delta \le \frac{F^*}{DKT} + \sum_{k=1}^K \frac{\mathbb{E}\text{Reg}_T(\boldsymbol{u}^k)}{DKT} + \sum_{k=1}^K \sum_{t=1}^T \frac{\mathbb{E}\langle \nabla F(\boldsymbol{w}_t^k) - \tilde{\boldsymbol{g}}_t^k, \Delta_t^k - \boldsymbol{u}^k\rangle}{DKT}.$$

*where $\boldsymbol{u}^k = -D\frac{\sum_{t=1}^T \nabla F(\boldsymbol{w}_t^k)}{\|\sum_{t=1}^T \nabla F(\boldsymbol{w}_t^k)\|}$.*

**Corollary 4.7.** *Suppose $F : \mathbb{R}^d \to \mathbb{R}$ is differentiable, L-Lipschitz, and $F(\boldsymbol{x}_1) - \inf_{\boldsymbol{x}} F(\boldsymbol{x}) \le F^*$, and suppose the domain of $\mathcal{A}$ is bounded by $D = \delta/T$. Then*

$$\mathbb{E}\|\nabla F(\overline{\boldsymbol{w}})\|_{2\delta} \le \frac{F^* + 2L\delta}{DKT} + \sum_{k=1}^K \frac{\mathbb{E}\text{Reg}_T(\hat{\boldsymbol{u}}^k)}{DKT} + \sum_{k=1}^K \sum_{t=1}^T \frac{\mathbb{E}\langle \nabla \hat{F}_\delta(\boldsymbol{w}_t^k) - \tilde{\boldsymbol{g}}_t^k, \Delta_t^k - \hat{\boldsymbol{u}}^k\rangle}{DKT}.$$

*where $\hat{\boldsymbol{u}}^k = -D\frac{\sum_{t=1}^T \nabla \hat{F}_\delta(\boldsymbol{w}_t^k)}{\|\sum_{t=1}^T \nabla \hat{F}_\delta(\boldsymbol{w}_t^k)\|}$.*

Next, we introduce a key result in our convergence analysis. At its core, Lemma 4.8 suggests that the variance of the non-private gradient oracle, defined as $\boldsymbol{g}_t^k$ in Algorithm 4, is approximately $O(dL^2/T)$. This result leads to the non-private term of $O(d\delta^{-1}\epsilon^{-3})$ in the data complexity, matching the optimal rate in non-private contexts. Together with Corollary 4.5, this lemma implies the main result as stated in Theorem 4.9.

**Lemma 4.8.** *Suppose $f(\boldsymbol{x}, z)$ is differentiable and L-Lipschitz in $\boldsymbol{x}$ and the domain of $\mathcal{A}$ is bounded by $D = \delta/T$, then the variance of the gradient oracle $\mathcal{O}$ (Algorithm 4) is bounded by*

$$\mathbb{E}\|\nabla \hat{F}_\delta(\boldsymbol{w}_t^k) - \boldsymbol{g}_t^k\|^2 \le \frac{16dL^2}{B_1} + \frac{64dL^2}{B_2 T}.$$

**Theorem 4.9.** *Suppose $f(\boldsymbol{x}, z)$ is differentiable and L-Lipschitz in $\boldsymbol{x}$, $F(\boldsymbol{x}_1) - \inf_{\boldsymbol{x}} F(\boldsymbol{x}) \le F^*$. Then for any $\delta, \epsilon > 0$ and $\alpha > 1, \rho > 0$, there exists*

$$M = \tilde{O}\left(d\left(\frac{F^*L^2}{\delta\epsilon^3} + \frac{L^3}{\epsilon^3}\right) + d^{3/2}\left(\frac{F^*L}{\rho\delta\epsilon^2} + \frac{L^2}{\rho\epsilon^2}\right)\right)$$

*such that upon running Algorithm 3 with projected OSD as the OCO algorithm, using Algorithm 4 as the private oracle, and setting $\sigma_t$ as defined in Corollary 4.4 and $B_1 = T + 1, B_2 = 1, D = \frac{\delta}{T}, T = \min\{(\frac{\sqrt{d}L\delta M}{F^*+L\delta})^{2/3}, (\frac{d^{3/2}L\delta M}{(F^*+L\delta)\rho})^{1/2}\}, K = \frac{M}{2T}$, the algorithm outputs an $(\alpha, \alpha\rho^2/2)$-RDP $(\delta, \epsilon)$-stationary point with $M$ data points.*

*Proof.* Since $B_1 = T+1, B_2 = 1$ satisfy $B_1 \ge TB_2/2$, Corollary 4.4 implies the privacy guarantee. For the convergence analysis, Corollary 4.7 states that

$$\mathbb{E}\|\nabla F(\overline{\boldsymbol{w}})\|_{2\delta} \le \frac{F^* + 2L\delta}{DKT} + \sum_{k=1}^K \frac{\mathbb{E}\text{Reg}_T(\hat{\boldsymbol{u}}^k)}{DKT} + \sum_{k=1}^K \sum_{t=1}^T \frac{\mathbb{E}\langle \nabla \hat{F}_\delta(\boldsymbol{w}_t^k) - \tilde{\boldsymbol{g}}_t^k, \Delta_t^k - \hat{\boldsymbol{u}}^k\rangle}{DKT}.$$

Recall that given a sequence of stochastic vectors $\boldsymbol{v}_1, \ldots, \boldsymbol{v}_T$, the regret of projected OSD is bounded by $\mathbb{E}[\mathrm{Reg}_T(\boldsymbol{u})] \leq D\sqrt{\sum_{t=1}^{T} \mathbb{E}\|\boldsymbol{v}_t\|^2}$ (Orabona, 2019). In our case, $\boldsymbol{v}_t = \tilde{\boldsymbol{g}}_t^k$. We can bound

$$\mathbb{E}\|\tilde{\boldsymbol{g}}_t^k\|^2 \leq 3\mathbb{E}\|\boldsymbol{g}_t^k - \nabla \hat{F}_\delta(\boldsymbol{w}_t^k)\|^2 + 3\mathbb{E}\|\nabla \hat{F}_\delta(\boldsymbol{w}_t^k)\|^2 + 3\mathbb{E}\|\mathrm{TREE}(t)\|^2$$
$$\leq 3 \cdot \frac{80 d L^2}{B_2 T} + 3L^2 + 3 \cdot \left(\frac{8\ln(T)d^{3/2}L}{B_2 T \rho}\right)^2,$$

where the first term is bounded by Lemma 4.8 with $B_1 \geq TB_2/2$, the second by Lipschitzness of $\hat{F}_\delta$ (Lemma 2.2 (i)), and the third by Corollary 4.5. Consequently, we bound the regret by

$$\mathbb{E}[\mathrm{Reg}_T(\hat{\boldsymbol{u}}^k)] \lesssim DL\sqrt{T}\left(\frac{\sqrt{d}}{\sqrt{B_2 T}} + 1 + \frac{d^{3/2}}{B_2 T \rho}\right). \tag{1}$$

Next, note that $\|\Delta_t^k - \hat{\boldsymbol{u}}^k\| \leq 2D$. Following the same previous bounds, we have

$$\mathbb{E}\langle \nabla \hat{F}_\delta(\boldsymbol{w}_t^k) - \tilde{\boldsymbol{g}}_t^k, \Delta_t^k - \hat{\boldsymbol{u}}^k\rangle \leq 2D\mathbb{E}[\|\nabla \hat{F}_\delta(\boldsymbol{w}_t^k) - \boldsymbol{g}_t^k\| + \|\mathrm{TREE}(t)\|]$$
$$\lesssim D\left(\frac{\sqrt{d}L}{\sqrt{B_2 T}} + \frac{d^{3/2}L}{B_2 T \rho}\right). \tag{2}$$

Upon substituting equation 1 and equation 2 into Corollary 4.7, we have

$$\mathbb{E}\|\nabla F(\overline{\boldsymbol{w}})\|_{2\delta} \lesssim \frac{F^* + L\delta}{DKT} + \frac{L}{\sqrt{T}}\left(\frac{\sqrt{d}}{\sqrt{B_2 T}} + 1 + \frac{d^{3/2}}{B_2 T \rho}\right) + \left(\frac{\sqrt{d}L}{\sqrt{B_2 T}} + \frac{d^{3/2}L}{B_2 T \rho}\right)$$

Upon setting $B_1 = T + 1, B_2 = 1, D = \frac{\delta}{T}$, the data size is $M = K(B_1 + B_2(T-1)) = 2KT$ and

$$\lesssim \frac{(F^* + L\delta)T}{\delta M} + \frac{\sqrt{d}L}{\sqrt{T}} + \frac{d^{3/2}L}{T\rho}$$

Upon setting $T = \min\{(\frac{\sqrt{d}L\delta M}{F^* + L\delta})^{2/3}, (\frac{d^{3/2}L\delta M}{(F^* + L\delta)\rho})^{1/2}\}$, we achieve the desired bound

$$\lesssim \left(\frac{F^* d L^2}{\delta M}\right)^{1/3} + \left(\frac{dL^3}{M}\right)^{1/3} + \left(\frac{F^* d^{3/2}L}{\rho\delta M}\right)^{1/2} + \left(\frac{d^{3/2}L^2}{\rho M}\right)^{1/2}.$$

Equivalently, the right hand side is bounded by $\epsilon$ if $M = \max\{\frac{F^* d L^2}{\delta\epsilon^3}, \frac{dL^3}{\epsilon^3}, \frac{F^* d^{3/2}L}{\rho\delta\epsilon^2}, \frac{d^{3/2}L^2}{\rho\epsilon^2}\}$. $\qquad\square$

## 5 CONCLUSION

This paper presents a novel zeroth-order algorithm for private nonsmooth nonconvex optimization. We prove that, given a dataset of size $M$, our algorithm finds a $(\delta, \epsilon)$-stationary point while achieving $(\alpha, \alpha\rho^2/2)$-RDP privacy guarantee so long as $M = \tilde{\Omega}\left(d(\frac{F^* L^2}{\delta\epsilon^3} + \frac{L^3}{\epsilon^3}) + d^{3/2}(\frac{F^* L}{\rho\delta\epsilon^2} + \frac{L^2}{\rho\epsilon^2})\right)$. Notably, this is the first algorithm for the private nonsmooth nonconvex optimization problem.

For future research, there are several intriguing directions. First, we are interested in exploring methods that could eliminate the need to sample $d$ uniform vectors per data point, a limitation we delve into in Section 3. Additionally, the design of efficient first-order private algorithms for nonsmooth nonconvex optimization stands out as an area of potential exploration. Finally, the optimality of the current rate is still uncertain, and finding the lower bounds remains an open problem.

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

# A PROOFS IN SECTION 2

## A.1 PROOF OF LEMMA 2.2

**Lemma 2.2.** *Suppose $h : \mathbb{R}^d \to \mathbb{R}$ is L-Lipschitz. Then (i) $\hat{h}_\delta$ is L-Lipschitz; (ii) $\|\hat{h}_\delta(\boldsymbol{x}) - h(\boldsymbol{x})\| \leq L\delta$; (iii) $\hat{h}_\delta$ is differentiable and $\frac{\sqrt{d}L}{\delta}$-smooth; (iv)*

$$\nabla \hat{h}_\delta(\boldsymbol{x}) = \mathbb{E}_{\boldsymbol{u} \sim \mathcal{U}_\mathbb{S}}[\tfrac{d}{\delta} h(\boldsymbol{x} + \delta\boldsymbol{u})\boldsymbol{u}] = \mathbb{E}_{\boldsymbol{u} \sim \mathcal{U}_\mathbb{S}}[\tfrac{d}{2\delta}(h(\boldsymbol{x} + \delta\boldsymbol{u}) - h(\boldsymbol{x} - \delta\boldsymbol{u}))\boldsymbol{u}];$$

*Proof.* For simplicity, we drop the subscript $\delta$ of $\hat{h}_\delta$. By Jensen's inequality and Lipschitzness,

$$\|\hat{h}(\boldsymbol{x}) - \hat{h}(\boldsymbol{y})\| \leq \mathbb{E}_{\boldsymbol{v}}\|h(\boldsymbol{x} + \delta\boldsymbol{v}) - h(\boldsymbol{y} + \delta\boldsymbol{v})\| \leq L\|\boldsymbol{x} - \boldsymbol{y}\|, \tag{i}$$

$$\|\hat{h}(\boldsymbol{x}) - h(\boldsymbol{x})\| \leq \mathbb{E}_{\boldsymbol{v}}\|h(\boldsymbol{x} + \delta\boldsymbol{v}) - h(\boldsymbol{x})\| \leq L\|\delta\boldsymbol{v}\| \leq L\delta. \tag{ii}$$

Next, since $h$ is Lipschitz, $h$ is almost every differentiable by Rademacher's theorem and the uniform smoothing $\hat{h}$ is also almost everywhere differentiable by (Bertsekas, 1973, Proposition 2.4). Moreover, it holds that $\nabla \hat{h}(\boldsymbol{x}) = \mathbb{E}_{\boldsymbol{v}}[\nabla h(\boldsymbol{x} + \delta\boldsymbol{v})]$. Denote $\triangle$ as the symmetric difference, i.e., $A \triangle B = (A \setminus B) \cup (B \setminus A)$, and let $S = B(\boldsymbol{x}, \delta) \triangle B(\boldsymbol{y}, \delta)$. Then

$$\|\nabla \hat{h}(\boldsymbol{x}) - \nabla \hat{h}(\boldsymbol{y})\| = \|\mathbb{E}_{\boldsymbol{v} \sim \mathcal{U}_\mathbb{B}}[\nabla h(\boldsymbol{x} + \delta\boldsymbol{v}) - \nabla h(\boldsymbol{y} + \delta\boldsymbol{v})]\|$$

$$= \left\| \int_{B(\boldsymbol{x},\delta)} \frac{\nabla h(\boldsymbol{v})}{\mathrm{vol}(B(\boldsymbol{x}, \delta))} \, d\boldsymbol{v} - \int_{B(\boldsymbol{y},\delta)} \frac{\nabla h(\boldsymbol{v})}{\mathrm{vol}(B(\boldsymbol{y}, \delta))} \, d\boldsymbol{v} \right\|$$

$$= \left\| \int_S \frac{\nabla h(\boldsymbol{v})}{\mathrm{vol}(\delta\mathbb{B})} \, d\boldsymbol{v} \right\| \leq \int_S \frac{\|\nabla h(\boldsymbol{v})\|}{\mathrm{vol}(\delta\mathbb{B})} \, d\boldsymbol{v} \leq \frac{\mathrm{vol}(S)L}{\mathrm{vol}(\delta\mathbb{B})} \leq \frac{\sqrt{d}L}{\delta}\|\boldsymbol{x} - \boldsymbol{y}\|,$$

where the second last inequality follows from $\|\nabla h(\boldsymbol{v})\| \leq L$, and the last follows from Lemma A.1.

Finally, by definition of expectation,

$$\mathbb{E}_{\boldsymbol{v} \sim \mathcal{U}_\mathbb{B}}[\nabla h(\boldsymbol{x} + \delta\boldsymbol{v})] = \frac{\int_\mathbb{B} \nabla_{\boldsymbol{x}} f(\boldsymbol{x} + \delta\boldsymbol{v}) \, d\boldsymbol{v}}{\mathrm{vol}(\mathbb{B})},$$

$$\mathbb{E}_{\boldsymbol{u} \sim \mathcal{U}_\mathbb{S}}[h(\boldsymbol{x} + \delta\boldsymbol{u})\boldsymbol{u}] = \frac{\int_\mathbb{S} f(\boldsymbol{x} + \delta\boldsymbol{u})\boldsymbol{u} \, d\boldsymbol{u}}{\mathrm{vol}(\mathbb{S})}.$$

By Stokes' theorem,

$$\int_\mathbb{B} \nabla_{\boldsymbol{x}} f(\boldsymbol{x} + \delta\boldsymbol{v}) \, d\boldsymbol{v} = \int_\mathbb{S} f(\boldsymbol{x} + \delta\boldsymbol{u})\boldsymbol{u} \, d\boldsymbol{u}$$

We then establish the first equality in (iv) by $\mathrm{vol}(\mathbb{B})/\mathrm{vol}(\mathbb{S}) = \delta/d$. Next, let $p$ be a Rademacher random variable and $\boldsymbol{u}' = \boldsymbol{u}/p$. Observe that $\boldsymbol{u}' \sim \mathcal{U}_\mathbb{S}$ as well. Therefore,

$$\mathbb{E}_{\boldsymbol{u} \sim \mathcal{U}_\mathbb{S}}[\tfrac{d}{\delta} h(\boldsymbol{x} + \delta\boldsymbol{u})\boldsymbol{u}] = \mathbb{E}_{p, \boldsymbol{u}'}[\tfrac{d}{\delta} h(\boldsymbol{x} + \delta p\boldsymbol{u}')p\boldsymbol{u}']$$

$$= \tfrac{1}{2}\mathbb{E}_{\boldsymbol{u}'}[\tfrac{d}{\delta} h(\boldsymbol{x} + \delta\boldsymbol{u})\boldsymbol{u} + \tfrac{d}{\delta} h(\boldsymbol{x} - \delta\boldsymbol{u})(-\boldsymbol{u})].$$

This proves the second equality in (iv). □

**Lemma A.1.** *Let $\boldsymbol{x}, \boldsymbol{y} \in \mathbb{R}^d$ and $S = B(\boldsymbol{x}, \delta) \triangle B(\boldsymbol{y}, \delta)$, then $\frac{\mathrm{vol}(S)}{\mathrm{vol}(\delta\mathbb{B})} \leq \frac{\sqrt{d}}{\delta}\|\boldsymbol{x} - \boldsymbol{y}\|$.*

*Proof.* Let $D = \|\boldsymbol{x} - \boldsymbol{y}\|$, $V_d(r)$ be the volume of an $d$-ball in $\mathbb{R}^d$, and $U_d(h, r)$ be the volume of the cap of an $d$-ball of height $h$. Observe that for $D \leq 2\delta$,

$$\frac{\mathrm{vol}(S)}{\mathrm{vol}(\delta\mathbb{B})} = 2 \cdot \frac{U_d(\delta + \frac{D}{2}, \delta) - U_d(\delta - \frac{D}{2}, \delta)}{V_d(\delta)}.$$

Next, we compute the value of $U_d(h, r)$.

$$U_d(h, r) = \int_0^h V_{d-1}\left(\sqrt{r^2 - (r - t)^2}\right) dt.$$

Recall that the volume of an $d$-ball is $V_d(r) = \frac{\pi^{d/2}}{\Gamma(d/2+1)} r^n$. Therefore,

$$\frac{\text{vol}(S)}{\text{vol}(\delta\mathbb{B})} = \frac{\frac{\pi^{(d-1)/2}}{\Gamma((d-1)/2+1)} \int_{\delta-\frac{D}{2}}^{\delta+\frac{D}{2}} (2\delta t - t^2)^{\frac{d-1}{2}} dt}{\frac{\pi^{d/2}}{\Gamma(d/2+1)} \delta^d}$$

$$= \frac{\Gamma(\frac{d}{2}+1)}{\sqrt{\pi}\Gamma(\frac{d-1}{2}+1)} \int_{\delta-\frac{D}{2}}^{\delta+\frac{D}{2}} \left(\frac{2t}{\delta} - \frac{t^2}{\delta^2}\right)^{\frac{d-1}{2}} \frac{1}{\delta} dt$$

$$= \frac{\Gamma(\frac{d}{2}+1)}{\sqrt{\pi}\Gamma(\frac{d-1}{2}+1)} \int_{1-\frac{D}{2\delta}}^{1+\frac{D}{2\delta}} (2u - u^2)^{\frac{d-1}{2}} du. \qquad (t = \delta u)$$

Note that $2u - u^2 \leq 1$ for all $u \in \mathbb{R}$, so the integral is bounded by $\int_{1-\frac{D}{2\delta}}^{1+\frac{D}{2\delta}} 1\, du = \frac{D}{\delta}$. The proof then follows from the fact that $\Gamma(\frac{d}{2}+1)/\Gamma(\frac{d-1}{2}+1) \leq \sqrt{2d}$ (proved in Lemma A.2 for completeness). $\qquad \square$

**Lemma A.2.** *For all $n \in \mathbb{N}$, $\Gamma(\frac{n}{2}+1)/\Gamma(\frac{n-1}{2}+1) \leq \sqrt{2n}$.*

*Proof.* Let $n!! = n(n-2)(n-4)\cdots = \prod_{k=0}^{\lceil \frac{n}{2}\rceil - 1}(n-2k)$, then by definition of $\Gamma(n+1) = n\gamma(n)$ and $\Gamma(\frac{1}{2}) = \sqrt{\pi}$, we have

$$\Gamma(\tfrac{n}{2}+1) = \begin{cases} \frac{n!!}{2^{n/2}}, & n \text{ is even} \\ \frac{\sqrt{\pi}n!!}{2^{(n+1)/2}}, & n \text{ is odd} \end{cases}$$

Consequently, we have

$$\frac{\Gamma(\frac{n}{2}+1)}{\Gamma(\frac{n-1}{2}+1)} = \begin{cases} \frac{n!!}{\sqrt{\pi}(n-1)!!}, & n \text{ is even} \\ \frac{\sqrt{\pi}n!!}{2(n-1)!!}, & n \text{ is odd} \end{cases} \leq \frac{n!!}{(n-1)!!}.$$

We finish the proof by induction. For $n = 1$, $\frac{1!!}{0!!} = 1 \leq \sqrt{2 \cdot 1}$. For $n = 2$, $\frac{2!!}{1!!} = 2 \leq \sqrt{2 \cdot 2}$. Now assume $\frac{m!!}{(m-1)!!} \leq \sqrt{2m}$ for all $m \leq n$. Then

$$\frac{(n+1)!!}{n!!} = \frac{n+1}{n}\frac{(n-1)!!}{(n-2)!!} \leq \frac{n+1}{n}\sqrt{2(n-1)} = \sqrt{2(n+1)}\frac{\sqrt{(n+1)(n-1)}}{n} \leq \sqrt{2(n+1)}.$$

This concludes the proof by induction. $\qquad \square$

## A.2 Proof of Corollary 2.3

**Corollary 2.3.** *Suppose $F : \mathbb{R}^d \to \mathbb{R}$ is $L$-Lipschitz. Then for any $\epsilon, \delta > 0$, $\|\nabla\hat{F}_\delta(\boldsymbol{x})\|_\delta \leq \epsilon$ implies that $\|\nabla F(\boldsymbol{x})\|_{2\delta} \leq \epsilon$.*

*Proof.* Recall that the $\delta$-subdifferential of $\hat{F}_\delta$ is defined as $\partial_\delta\hat{F}_\delta(\boldsymbol{x}) = \text{conv}(\cup_{\boldsymbol{y}\in B(\boldsymbol{x},\delta)} \nabla\hat{F}_\delta(\boldsymbol{y}))$. In other words, for each $\boldsymbol{g} \in \partial_\delta\hat{F}_\delta(\boldsymbol{x})$, $\boldsymbol{g}$ is of form of $\boldsymbol{g} = \sum \lambda_i \nabla\hat{F}_\delta(\boldsymbol{y}_i)$ where $\lambda_i$'s are convex coefficients and $\boldsymbol{y}_i \in B(\boldsymbol{x}, \delta)$. (Lin et al., 2022, Theorem 10) has proved that $\nabla\hat{F}_\delta(\boldsymbol{y}_i) \in \partial_\delta F(\boldsymbol{y}_i)$. In addition, since $\|\boldsymbol{y}_i - \boldsymbol{x}\| \leq \delta$, we have $\partial_\delta F(\boldsymbol{y}_i) \subset \partial_{2\delta} F(\boldsymbol{x})$. Consequently, we have

$$\nabla\hat{F}_\delta(\boldsymbol{y}_i) \in \partial_\delta F(\boldsymbol{y}_i) \subset \partial_{2\delta} F(\boldsymbol{x})$$

and thus $\boldsymbol{g} \in \partial_{2\delta} F(\boldsymbol{x})$ for all $\boldsymbol{g} \in \partial_\delta\hat{F}_\delta(\boldsymbol{x})$. We conclude the proof by recalling Definition 2.1 that

$$\|\nabla\hat{F}_\delta(\boldsymbol{x})\|_\delta = \inf\{\|\boldsymbol{g}\| : \boldsymbol{g} \in \partial_\delta\hat{F}_\delta(\boldsymbol{x})\}, \quad \|\nabla F(\boldsymbol{x})\|_{2\delta} = \inf\{\|\boldsymbol{g}\| : \boldsymbol{g} \in \partial_{2\delta} F(\boldsymbol{x})\}$$

and the previous result that $\|\partial_\delta\hat{F}_\delta(\boldsymbol{x}) \subset \partial_{2\delta} F(\boldsymbol{x})$. $\qquad \square$

## B  PROOFS IN SECTION 3

**Lemma 3.1.** *If $f(\boldsymbol{x}, z)$ is differentiable and L-Lipschitz in $\boldsymbol{x}$, then for any $\delta > 0, \boldsymbol{x} \in \mathbb{R}^d$ and neighboring data batches $z_{1:b}, z'_{1:b}$ of size $b$,*

$$\mathbb{E}[\text{GRAD}_{f,\delta}(\boldsymbol{x}, z_{1:b})] = \nabla \hat{F}_\delta(\boldsymbol{x}), \qquad \text{(unbiased)}$$

$$\mathbb{E}\|\text{GRAD}_{f,\delta}(\boldsymbol{x}, z_{1:b}) - \nabla \hat{F}_\delta(\boldsymbol{x})\|^2 \le \tfrac{16dL^2}{b}, \qquad \text{(variance)}$$

$$\|\text{GRAD}_{f,\delta}(\boldsymbol{x}, z_{1:b}) - \text{GRAD}_{f,\delta}(\boldsymbol{x}, z'_{1:b})\| \le \tfrac{2dL}{b}. \qquad \text{(sensitivity)}$$

*Proof.* For simplicity, let $\boldsymbol{g}_{ij} = \frac{d}{2\delta}(f(\boldsymbol{x}+\delta\boldsymbol{u}_{ij}, z_i) - f(\boldsymbol{x}-\delta\boldsymbol{u}_{ij}, z_i))\boldsymbol{u}_{ij}$ so that $\text{GRAD}_{f,\delta}(\boldsymbol{x}, z_{1:b}) = \frac{1}{b}\sum_{i=1}^b \frac{1}{d}\sum_{j=1}^d \boldsymbol{g}_{ij}$. Since $f$ is $L$-Lipschitz, it follows that $\|\boldsymbol{g}_{ij}\| \le dL$.

First, by $F(\boldsymbol{x}) = \mathbb{E}_z[f(\boldsymbol{x}, z)]$ and Lemma 2.2 (iv),

$$\mathbb{E}_{\boldsymbol{u},z}[\boldsymbol{g}_{ij}] = \mathbb{E}_{\boldsymbol{u}}[\tfrac{d}{2\delta}(F(\boldsymbol{x}+\delta\boldsymbol{u}_{ij}) - F(\boldsymbol{x}-\delta\boldsymbol{u}_{ij}))\boldsymbol{u}_{ij}] = \nabla \hat{F}_\delta(\boldsymbol{x}).$$

Taking the average over $i, j$ gives $\mathbb{E}[\text{GRAD}_{f,\delta}(\boldsymbol{x}, z_{1:b})] = \nabla \hat{F}_\delta(\boldsymbol{x})$.

Next, let $\overline{\boldsymbol{g}}_i = \frac{1}{d}\sum_{j=1}^d \boldsymbol{g}_{ij}$ and observe that $\mathbb{E}[\overline{\boldsymbol{g}}_i] = \nabla \hat{F}_\delta(\boldsymbol{x})$. Since $\overline{\boldsymbol{g}}_i - \nabla \hat{F}_\delta(\boldsymbol{x})$ are mean-zero and independent for all $i$, we have

$$\mathbb{E}[\langle \overline{\boldsymbol{g}}_i - \nabla \hat{F}_\delta(\boldsymbol{x}), \overline{\boldsymbol{g}}_{i'} - \nabla \hat{F}_\delta(\boldsymbol{x})\rangle] = \mathbb{E}_{z_{i'}}[\langle \mathbb{E}_{z_i}[\overline{\boldsymbol{g}}_i - \nabla \hat{F}_\delta(\boldsymbol{x})], \overline{\boldsymbol{g}}_{i'} - \nabla \hat{F}_\delta(\boldsymbol{x})\rangle | z_{i'}] = 0.$$

In other words, all cross-terms are 0. Therefore,

$$\mathbb{E}\|\text{GRAD}_{f,\delta}(\boldsymbol{x}, z_{1:b}) - \nabla \hat{F}_\delta(\boldsymbol{x})\|^2$$
$$= \tfrac{1}{b^2}\sum_{i=1}^b \mathbb{E}\|\overline{\boldsymbol{g}}_i - \nabla \hat{F}_\delta(\boldsymbol{x})\|^2$$
$$\le \tfrac{1}{b^2}\sum_{i=1}^b 2\mathbb{E}\|\overline{\boldsymbol{g}}_i - \nabla \hat{f}_\delta(\boldsymbol{x}, z_i)\|^2 + 2\mathbb{E}\|\nabla \hat{f}(\boldsymbol{x}, z_i) - \nabla \hat{F}_\delta(\boldsymbol{x})\|^2. \qquad (3)$$

We first evaluate the second term. By Lemma 2.2 (i), $\hat{f}_\delta(\cdot, z_i)$ and $F_\delta$ are $L$-Lipschitz, which implies that $\|\nabla \hat{f}_\delta(\boldsymbol{x}, z_i)\| \le L$ and $\|\nabla \hat{F}_\delta(\boldsymbol{x})\| \le L$. Consequently, $\mathbb{E}\|\nabla \hat{f}(\boldsymbol{x}, z_i) - \nabla \hat{F}_\delta(\boldsymbol{x})\|^2 \le (2L)^2$. Next, note that $\boldsymbol{g}_{ij} - \nabla \hat{f}_\delta(\boldsymbol{x}, z_i)|z_i$ are mean-zero and independent for all $j$. Therefore,

$$\mathbb{E}\|\overline{\boldsymbol{g}}_i - \nabla \hat{f}_\delta(\boldsymbol{x}, z_i)\|^2 = \tfrac{1}{d^2}\sum_{j=1}^d \mathbb{E}\|\boldsymbol{g}_{ij} - \nabla \hat{f}_\delta(\boldsymbol{x}, z_i)\|^2 \le \tfrac{(d+1)^2 L^2}{d}.$$

The last inequality follows from $\|\boldsymbol{g}_{ij} - \nabla \hat{f}_\delta(\boldsymbol{x}, z_i)\| \le \|\boldsymbol{g}_{ij}\| + \|\nabla \hat{f}_\delta(\boldsymbol{x}, z_i)\| \le (d+1)L$. Upon substituting back into equation 3, we have

$$\mathbb{E}\|\text{GRAD}_{f,\delta}(\boldsymbol{x}, z_{1:b}) - \nabla \hat{F}_\delta(\boldsymbol{x})\|^2 \le \tfrac{2}{b}\left(\tfrac{(d+1)^2 L^2}{d} + (2L)^2\right) \le \tfrac{16dL^2}{b}.$$

Finally, let $\boldsymbol{g}'_{ij}, \overline{\boldsymbol{g}}'_i$ be defined in the same way as $\boldsymbol{g}_{ij}, \overline{\boldsymbol{g}}_i$ but using $z'_{1:b}$. Since $z_{1:b}, z'_{1:b}$ are neighboring, $\overline{\boldsymbol{g}}_i - \overline{\boldsymbol{g}}'_i = 0$ for all but one $i$. Let $k$ be the index of the different data point. Then

$$\|\text{GRAD}_{f,\delta}(\boldsymbol{x}, z_{1:b}) - \text{GRAD}_{f,\delta}(\boldsymbol{x}, z'_{1:b})\| = \tfrac{1}{b}\|\overline{\boldsymbol{g}}_k - \overline{\boldsymbol{g}}'_k\| \le \tfrac{1}{bd}\sum_{j=1}^d \|\boldsymbol{g}_{kj} - \boldsymbol{g}'_{kj}\| \le \tfrac{2dL}{b}. \quad \square$$

**Lemma 3.2.** *If $f(\boldsymbol{x}, z)$ is differentiable and L-Lipschitz in $\boldsymbol{x}$, then for any $\delta > 0$, $\boldsymbol{x}, \boldsymbol{y} \in \mathbb{R}^d$ and neighboring data batches $z_{1:b}, z'_{1:b}$ of size $b$,*

$$\mathbb{E}[\text{DIFF}_{f,\delta}(\boldsymbol{x}, \boldsymbol{y}, z_{1:b})] = \nabla \hat{F}_\delta(\boldsymbol{x}) - \nabla \hat{F}_\delta(\boldsymbol{y}), \qquad \text{(unbiased)}$$

$$\mathbb{E}\|\text{DIFF}_{f,\delta}(\boldsymbol{x}, \boldsymbol{y}, z_{1:b}) - [\nabla \hat{F}_\delta(\boldsymbol{x}) - \nabla \hat{F}_\delta(\boldsymbol{y})]\|^2 \le \tfrac{16dL^2}{b\delta^2}\|\boldsymbol{x} - \boldsymbol{y}\|^2, \qquad \text{(variance)}$$

$$\|\text{DIFF}_f(\boldsymbol{x}, \boldsymbol{y}, z_{1:b}) - \text{DIFF}_f(\boldsymbol{x}, \boldsymbol{y}, z'_{1:b})\| \le \tfrac{2dL}{b\delta}\|\boldsymbol{x} - \boldsymbol{y}\|. \qquad \text{(sensitivity)}$$

*Proof.* Let $\boldsymbol{d}_{ij} = \frac{d}{\delta}(f(\boldsymbol{x}+\delta\boldsymbol{u}_{ij}, z_i) - f(\boldsymbol{y}+\delta\boldsymbol{u}_{ij}, z_i))\boldsymbol{u}_{ij}$. Then it follows that $\text{DIFF}_{f,\delta}(\boldsymbol{x}, \boldsymbol{y}, z_{1:b}) = \frac{1}{b}\sum_{i=1}^b \frac{1}{d}\sum_{j=1}^d \boldsymbol{d}_{ij}$. Since $f$ is $L$-Lipschitz, it also holds that $\|\boldsymbol{d}_{ij}\| \le \frac{dL}{\delta}\|\boldsymbol{x} - \boldsymbol{y}\|$.

First, by $F(\boldsymbol{x}) = \mathbb{E}_z[f(\boldsymbol{x}, z)]$ and Lemma 2.2 (iv),

$$\mathbb{E}_{\boldsymbol{u},z}[\boldsymbol{d}_{ij}] = \mathbb{E}_{\boldsymbol{u}}[\tfrac{d}{\delta}(F(\boldsymbol{x}+\delta\boldsymbol{u}_{ij}) - F(\boldsymbol{y}+\delta\boldsymbol{u}_{ij})\boldsymbol{u}_{ij}] = \nabla \hat{F}_\delta(\boldsymbol{x}) - \nabla \hat{F}_\delta(\boldsymbol{y}).$$

Taking the average over $i, j$ gives $\mathbb{E}_{\boldsymbol{u}, z}[\text{DIFF}_{f, \delta}(\boldsymbol{x}, \boldsymbol{y}, z_{1:b})] = \nabla \hat{F}_\delta(\boldsymbol{x}) - \nabla \hat{F}_\delta(\boldsymbol{y})$.

Next, let $\overline{\boldsymbol{d}}_i = \frac{1}{k} \sum_{j=1}^k \boldsymbol{d}_{ij}$, $\Delta_F = \nabla \hat{F}_\delta(\boldsymbol{x}) - \hat{F}_\delta(\boldsymbol{y})$, and $\Delta_f(z_i) = \nabla \hat{f}_\delta(\boldsymbol{x}, z_i) - \nabla \hat{f}_\delta(\boldsymbol{y}, z_i)$. Observe that $\mathbb{E}[\overline{\boldsymbol{d}}_i] = \Delta_F$ and $\mathbb{E}[\boldsymbol{d}_{ij}|z_i] = \Delta_f(z_i)$. Since $\overline{\boldsymbol{d}}_i - \Delta_F$ are independent and mean-zero for all $i$, we have

$$\mathbb{E}\|\text{DIFF}_{f, \delta}(\boldsymbol{x}, \boldsymbol{y}, z_{1:b}) - [\nabla \hat{F}_\delta(\boldsymbol{x}) - \nabla \hat{F}_\delta(\boldsymbol{y})]\|^2$$
$$= \frac{1}{b^2} \sum_{i=1}^b \mathbb{E}\|\overline{\boldsymbol{d}}_i - \Delta_F\|^2$$
$$\leq \frac{1}{b^2} \sum_{i=1}^b 2\mathbb{E}\|\overline{\boldsymbol{d}}_i - \Delta_f(z_i)\|^2 + 2\mathbb{E}\|\Delta_f(z_i) - \Delta_F\|^2. \qquad (4)$$

We first evaluate the second term. By Lemma 2.2 (iii), $\hat{f}_\delta, F_\delta$ are $\frac{\sqrt{d}L}{\delta}$-smooth, which implies that $\|\Delta_f(z_i)\|, \|\Delta_F\| \leq \frac{\sqrt{d}L}{\delta}\|\boldsymbol{x} - \boldsymbol{y}\|$. Consequently, $\mathbb{E}\|\Delta_f(z_i) - \Delta_F\|^2 \leq (\frac{2\sqrt{d}L}{\delta}\|\boldsymbol{x} - \boldsymbol{y}\|)^2$. Next, since $\boldsymbol{d}_{ij} - \Delta_f(z_i)|z_i$ are mean-zero and independent for all $j$,

$$\mathbb{E}\|\overline{\boldsymbol{d}}_i - \Delta_f(z_i)\|^2 \leq \frac{1}{d^2} \sum_{j=1}^d \mathbb{E}\|\boldsymbol{d}_{ij} - \Delta_f(z_i)\|^2 \leq \frac{(2d^2 + 2d)L^2}{d\delta^2}\|\boldsymbol{x} - \boldsymbol{y}\|^2.$$

The last inequality follows from $\|\boldsymbol{d}_{ij}\| \leq \frac{dL}{b}\|\boldsymbol{x} - \boldsymbol{y}\|$ and $\|\Delta_f(z_i)\| \leq \frac{\sqrt{d}L}{\delta}\|\boldsymbol{x} - \boldsymbol{y}\|$. Therefore, upon substituting back into equation 4, we have

$$\mathbb{E}\|\text{DIFF}_{f, \delta}(\boldsymbol{x}, \boldsymbol{y}, z_{1:b}) - [\nabla \hat{F}_\delta(\boldsymbol{x}) - \nabla \hat{F}_\delta(\boldsymbol{y})]\|^2 \leq \frac{16dL^2}{b\delta^2}\|\boldsymbol{x} - \boldsymbol{y}\|^2.$$

Finally, let $\boldsymbol{d}'_{ij}, \overline{\boldsymbol{d}}'_i$ be defined in the same way as $\boldsymbol{d}_{ij}, \overline{\boldsymbol{d}}_i$ but using $z'_{1:b}$. Since $z_{1:b}, z'_{1:b}$ are neighboring, $\overline{\boldsymbol{d}}_i - \overline{\boldsymbol{d}}'_i = 0$ for all but one $i$. Let $k$ be the index of the different data point. Then

$$\|\text{DIFF}_f(\boldsymbol{x}, \boldsymbol{y}, z_{1:b}) - \text{DIFF}_f(\boldsymbol{x}, \boldsymbol{y}, z'_{1:b})\| = \frac{1}{b}\|\overline{\boldsymbol{d}}_k - \overline{\boldsymbol{d}}'_k\| \leq \frac{2dL}{b\delta}\|\boldsymbol{x} - \boldsymbol{y}\|. \qquad \square$$

## C  PROOFS IN SECTION 4.2

### C.1  PROOF OF THEOREM 4.3

Before we prove the theorem, we first prove a more general composition of RDP mechanisms. For two datasets $Z, Z' \in \mathcal{Z}$, we denote $Z \simeq_q Z'$ if $Z, Z'$ are neighbors and they differ at the $q$-th entry.

**Lemma C.1.** *For any domain $\mathcal{Z}$, let $\mathcal{R}^{(i)} : \mathcal{S}^{(1)} \times \cdots \times \mathcal{S}^{(i-1)} \times \mathcal{Z} \to \mathcal{S}^{(i)}$ be a sequence of algorithms such that $\mathcal{R}^{(i)}(\boldsymbol{x}_{1:i-1}, \cdot)$ is $(\alpha, \epsilon_i)$-RDP for all auxiliary inputs $\boldsymbol{x}_{1:i-1} \in \mathcal{S}^{(i)} \times \cdots \times \mathcal{S}^{(i-1)}$. Let $f_{\mathcal{R}^{(i)}(\boldsymbol{x}_{1:i-1}, Z)}$ be the distribution of $\mathcal{R}^{(i)}(\boldsymbol{x}_{1:i-1}, Z)$. Suppose each dataset $Z \in \mathcal{Z}$ has $m$ data, and for each $q \in [m]$, let*

$$\text{IN}(q) := \{i \in [n] : f_{\mathcal{R}^{(i)}(\cdot, Z)} = f_{\mathcal{R}^{(i)}(\cdot, Z')}, \forall Z \simeq_q Z'\}, \quad \text{OUT}(q) := [n] \setminus \text{IN}(q).$$

*Let $\mathcal{A}_n : \mathcal{Z} \to \mathcal{S}^{(1)} \times \cdots \times \mathcal{S}^{(n)}$ be the algorithm that given a dataset $Z \in \mathcal{Z}$ sequentially computes $\boldsymbol{x}_i = \mathcal{R}^{(i)}(\boldsymbol{x}_{1:i-1}, z_i)$ for $i \in [n]$ and then outputs $\boldsymbol{x}_{1:n}$. Then $\mathcal{A}_n$ is $(\alpha, \epsilon)$-RDP, where*

$$\epsilon \leq \max_{q \in [m]} \sum_{i \in \text{OUT}(q)} \epsilon_i.$$

*Proof.* Suppose $Z \simeq_q Z'$ and let $P, Q$ be the distributions of $\mathcal{A}_n(Z), \mathcal{A}_n(Z')$ respectively. Note that $P = P_1 \times \cdots \times P_n$ where $P_i$ is the distribution of $\mathcal{A}_i(Z)$ given $\mathcal{A}_{i-1}(Z)$, i.e., $P_i(\boldsymbol{x}_i|\boldsymbol{x}_{1:i-1}) = f_{\mathcal{R}^{(i)}(\boldsymbol{x}_{1:i-1}, Z)}(\boldsymbol{x}_i)$. Similarly, $Q = Q_1 \times \cdots \times Q_n$ where $Q_i(\boldsymbol{x}_i|\boldsymbol{x}_{1:i-1}) = f_{\mathcal{R}^{(i)}(\boldsymbol{x}_{1:i-1}, Z')}(\boldsymbol{x}_i)$.

For all $i \in \text{IN}(q)$, $P_i(\cdot|\boldsymbol{x}_{1:i-1}) = Q_i(\cdot|\boldsymbol{x}_{1:i-1})$ for all $\boldsymbol{x}_{1:i-1}$ by definition of $\text{IN}(q)$; and for all $i \in \text{OUT}(q)$, since $\mathcal{R}^{(i)}(\boldsymbol{x}_{1:i-1}, \cdot)$ is $(\alpha, \epsilon_i)$-RDP, $D_\alpha(P_i(\cdot|\boldsymbol{x}_{1:i-1})\|Q_i(\cdot|\boldsymbol{x}_{1:i-1}) \leq \epsilon_i$. Therefore,

$$\int P_i(\boldsymbol{x}_i|\boldsymbol{x}_{1:i-1})^\alpha Q_i(\boldsymbol{x}_i|\boldsymbol{x}_{1:i-1})^{1-\alpha} \, d\boldsymbol{x}_i$$

$$= \exp((\alpha - 1)D_\alpha(P_i(\cdot|\boldsymbol{x}_{1:i-1})\|Q_i(\cdot|\boldsymbol{x}_{1:i-1})) \leq \begin{cases} 1, & i \in \text{IN}(q), \\ e^{(\alpha-1)\epsilon_i}, & i \in \text{OUT}(q). \end{cases}$$

We then conclude the proof by recursively integrating from $d\boldsymbol{x}_n$ to $d\boldsymbol{x}_1$:

$$e^{(\alpha-1)D_\alpha(P\|Q)} = \int \prod P_i(\boldsymbol{x}_i|\boldsymbol{x}_{1:i-1})^\alpha Q_i(\boldsymbol{x}_i|\boldsymbol{x}_{1:i-1})^{1-\alpha} \, d\boldsymbol{x}_n \cdots d\boldsymbol{x}_1 \leq \prod_{i \in \text{OUT}(q)} e^{(\alpha-1)\epsilon_i}. \quad \square$$

Next, we introduce several tree notations. Let $k \in \mathbb{N}_0$ and consider a complete binary tree with $n = 2^k$ leaves. Leaves are indexed by $i \in [n]$ (or $(i,i)$ interchangeably) in an ascending order from left to right, and every other node is indexed by $(i,j)$ where $i$ is its left-most child leaf and $j$ is its right-most child leaf. Let $\mathbb{T}$ be the set of all nodes, and let $<$ be a total order on $\mathbb{T}$ such that $(a,b) < (c,d)$ if either $b < d$ or $b = d, a < c$. In other words, we compare rightmost child first and break tie with leftmost child. We assume $(\mathbb{T}, <)$ is sorted in ascending order, then it's valid to label a sequence w.r.t. $\mathbb{T}$ (e.g., we can rewrite $a_1, \ldots, a_{|\mathbb{T}|}$ as $a_{(i,j)}$ for $(i,j) \in \mathbb{T}$). We denote $a_{(c,d)<(a,b)}$ as a short-hand notation of $\{a_{(c,d)} : (c,d) < (a,b)\}$. Now we are ready to prove the theorem.

**Theorem 4.3.** *Let $\mathcal{X}$ be state space and $\mathcal{Z}^{(1)}, \ldots, \mathcal{Z}^{(n)}$ be dataset spaces, and denote $\mathcal{Z}^{(1:i)} = \mathcal{Z}^{(1)} \times \cdots \times \mathcal{Z}^{(i)}$. Let $\mathcal{M}^{(i)} : \mathcal{X}^{i-1} \times \mathcal{Z}^{(i)} \to \mathcal{X}$ be a sequence of algorithms for $i \in [n]$, and let $\mathcal{A} : \mathcal{Z}^{(1:n)} \to \mathcal{X}^n$ be the algorithm that, given a dataset $Z_{1:n} \in \mathcal{Z}^{(1:n)}$, sequentially computes $\boldsymbol{x}_i = \sum_{j=1}^{i} \mathcal{M}^{(j)}(\boldsymbol{x}_{1:j-1}, Z_i) + \mathrm{TREE}(i)$ for $i \in [n]$ and then outputs $\boldsymbol{x}_{1:n}$. Suppose for all $i \in [n]$ and neighboring $Z_{1:n}, Z'_{1:n} \in \mathcal{Z}^{(1:n)}$, $\|\mathcal{M}^{(i)}(\boldsymbol{x}_{1:i-1}, Z_i) - \mathcal{M}^{(i)}(\boldsymbol{x}_{1:i-1}, Z'_i)\| \leq s_i$ for all auxiliary inputs $\boldsymbol{x}_{1:i-1} \in \mathcal{X}^{i-1}$. Then for all $\alpha > 1$, $\mathcal{A}$ is $(\alpha, \alpha\rho^2/2)$-RDP where*

$$\rho \leq \sqrt{2 \ln n} \cdot \max_{b \in [n], i \leq b} \frac{s_i}{\sigma_b}.$$

*Proof.* For any $n \in \mathbb{N}$, define $\mathbb{T}$ as he tree set with $2^{\lceil \log_2 n \rceil}$ leaves. Let $\mathcal{Z} = \mathcal{Z}^{(1:n)}$ and $S^{(a,b)} = \mathcal{X}$ for all $(a,b) \in \mathbb{T}$. For each $(a,b) \in \mathbb{T}$, define $\mathcal{R}^{(a,b)} : (\prod_{(c,d)<(a,b)} S^{(c,d)}) \times \mathcal{Z} \to S^{(a,b)}$ as

$$\mathcal{R}^{(a,b)}(\boldsymbol{y}_{(c,d)<(a,b)}, Z_{1:n}) = \sum_{i=a}^{b} \mathcal{M}^{(i)}(\boldsymbol{x}_{1:i-1}, Z_i) + \xi_b, \tag{5}$$

where $\boldsymbol{x}_j = \sum_{(c,d) \in \mathrm{NODE}(j)} \boldsymbol{y}_{(c,d)}$ for each $j \in [i-1]$ and $\xi_b \sim N(0, \sigma_b^2 I)$. We can check that $\boldsymbol{x}_j$ is well-defined: for all $j \leq b$ and for all $(c,d) \in \mathrm{NODE}(j)$, we have $(c,d) < (a,b)$ by definition of $\mathrm{NODE}$ and thus $\boldsymbol{x}_{1:i-1}$ can be computed from $\boldsymbol{y}_{(i,j)<(a,b)}$ for all $i \in [a,b]$.

Next, since for all neighboring $Z_{1:n}, Z'_{1:n} \in \mathcal{Z}$, $\|\mathcal{M}^{(i)}(\boldsymbol{x}_{1:i-1}, Z_i) - \mathcal{M}^{(i)}(\boldsymbol{x}_{1:i-1}, Z'_i)\| \leq s_i$, $\mathcal{R}^{(a,b)}(\boldsymbol{y}_{(c,d)<(a,b)}, \cdot)$ has sensitivity bounded by $s_{(a,b)} := \max_{i \in [a,b]} s_i$. Therefore, upon adding Gaussian noise $\mathcal{N}(0, \sigma_b^2 I)$, $\mathcal{R}^{(a,b)}(\boldsymbol{y}_{(c,d)<(a,b)}, \cdot)$ is $(\alpha, \epsilon_{(a,b)})$-RDP where $\epsilon_{(a,b)} = \alpha s_{(a,b)}^2/2\sigma_b^2$.

Now we are ready to apply Lemma C.1. Let $\mathcal{A}_\mathcal{R} : \mathcal{Z} \to \prod S^{(a,b)}$ be the composition of $\mathcal{R}$, i.e., given dataset $Z_{1:n}$ sequentially computes $\boldsymbol{y}_{(a,b)} = \mathcal{R}^{(a,b)}(\boldsymbol{y}_{(c,d)<(a,b)}, Z_{1:n})$ and outputs $\boldsymbol{y}_{(a,b) \in \mathbb{T}}$. Then by Lemma C.1, $\mathcal{A}_\mathcal{R}$ is $(\alpha, \epsilon)$-RDP where

$$\epsilon \leq \max_q \sum_{(a,b) \in \mathrm{OUT}(q)} \epsilon_{(a,b)} = \max_q \sum_{(a,b) \in \mathrm{OUT}(q)} \max_{i \in [a,b]} \frac{\alpha s_i^2}{2\sigma_b^2}.$$

Observe that node $(a,b) \in \mathrm{OUT}(q)$ if and only if the $q$-th data in $Z_{1:n}$ is in $Z_i$ and $i \in [a,b]$. Since there is exactly one such node in each of $\lceil \log_2 n \rceil + 1$ layers in the binary tree associated with $\mathbb{T}$, we have $|\mathrm{OUT}(q)| = \lceil \log_2 n \rceil + 1 \leq 2 \ln n$. Consequently,

$$\epsilon = \frac{\alpha\rho^2}{2} \leq \alpha \ln n \cdot \max_{b \in [n], i \leq b} \frac{s_i^2}{\sigma_b^2}.$$

Finally, we conclude the proof by claiming that algorithm $\mathcal{A}$ defined in the theorem can be derived from $\mathcal{A}_\mathcal{R}$ after post-processing. Observe that $\sum_{(a,b) \in \mathrm{NODE}(i)} \sum_{j=a}^{b} = \sum_{j=1}^{i}$, so by equation 5,

$$\boldsymbol{x}_i = \sum_{(a,b) \in \mathrm{NODE}(i)} \boldsymbol{y}_{(a,b)} = \sum_{(a,b) \in \mathrm{NODE}(i)} \left( \sum_{j=a}^{b} \mathcal{M}^{(j)}(\boldsymbol{x}_{1:j-1}, Z_j) + \xi_b \right)$$

$$= \sum_{j=1}^{i} \mathcal{M}^{(j)}(\boldsymbol{x}_{1:j-1}, Z_j) + \mathrm{TREE}(i).$$

Therefore, $\mathcal{A}$ is indeed a post-processing of $\mathcal{A}_\mathcal{R}$, which concludes the proof. $\square$

## C.2 PROOF OF COROLLARY 4.4

**Corollary 4.4.** *Suppose $f(\boldsymbol{x}, z)$ is differentiable and $L$-Lipschitz in $\boldsymbol{x}$ and the domain of $\mathcal{A}$ is bounded by $D = \delta/T$. Let $B_1, B_2$ satisfies $B_1 \geq TB_2/2$. Then for any $\alpha > 1, \rho > 0$, Algorithm 3 is $(\alpha, \alpha\rho^2/2)$-RDP if we set the noises $\sigma_{1:T}$ in the tree mechanism as*

$$\sigma_t = \sigma := \frac{\sqrt{2\ln T}\, 4dL}{B_2 T\rho}.$$

*Proof.* Since the private oracle defined in Algorithm 4 uses disjoint data in different iteration $k \in [K]$, Algorithm 3 is a post-processing of $\{(\tilde{\boldsymbol{g}}_1^k, \ldots, \tilde{\boldsymbol{g}}_T^k)\}_{k\in[K]}$. Therefore, it suffices to prove that for each $k \in [K]$, the composition $(\tilde{\boldsymbol{g}}_1^k, \ldots, \tilde{\boldsymbol{g}}_T^k)$ is $(\alpha, \alpha\rho^2/2)$-RDP.

The key is to observe that $(\tilde{\boldsymbol{g}}_1^k, \ldots, \tilde{\boldsymbol{g}}_T^k)$ can be written as the adaptive composition in Theorem 4.3. Let $\boldsymbol{g}_1^k = \mathcal{M}^{(1)}(Z_1^k) \triangleq \text{GRAD}_{f,\delta}(\boldsymbol{w}_1^k, Z_1^k)$ and $\boldsymbol{d}_i^k = \mathcal{M}^{(i)}(\tilde{\boldsymbol{g}}_{1:i-1}^k, Z_i^k) \triangleq \text{DIFF}_{f,\delta}(\boldsymbol{w}_i^k, \boldsymbol{w}_{i-1}^k, Z_i^k)$. This is well-defined because $\boldsymbol{w}_i^k$ is a post-processing of $\tilde{\boldsymbol{g}}_{1:i-1}^k$ by construction of O2NC. Therefore, $\boldsymbol{g}_t^k = \boldsymbol{g}_1^k + \sum_{i=2}^t \boldsymbol{d}_i^k = \sum_{i=1}^t \mathcal{M}^{(i)}(\tilde{\boldsymbol{g}}_{1:i-1}^k, Z_i^k)$, and Theorem 4.3 can be applied.

By Lemma 3.1, the sensitivity of $\mathcal{M}^{(1)}$ is bounded by $s_1 = \frac{2dL}{B_1}$. By Remark 4.1 and Lemma 3.2, the sensitivity of $\mathcal{M}^{(i)}(\tilde{\boldsymbol{g}}_{1:i-1}^k, \cdot)$ is bounded by $s_i = \frac{2dL}{B_2\delta}\|\boldsymbol{w}_i^k - \boldsymbol{w}_{i-1}^k\| \leq \frac{4dL}{B_2 T}$ for all $i \geq 2$. Since we assume $B_1 \geq TB_2/2$, it holds that $s_1 \leq s_t$ for all $t \geq 2$. Therefore, upon setting $\sigma_t = \sqrt{2\ln T}\, s_2/\rho$ for all $t \in [T]$, Theorem 4.3 implies that $(\tilde{\boldsymbol{g}}_1^k, \ldots \tilde{\boldsymbol{g}}_T^k)$ is $(\alpha, \alpha\rho'^2/2)$-RDP where

$$\rho' \leq \sqrt{2\ln T} \cdot \max_{b\in[n], i\leq n} \frac{s_i}{\sigma_b} \leq \sqrt{2\ln T} \cdot \frac{s_2}{\sigma_2} = \rho. \qquad \square$$

## C.3 PROOF OF COROLLARY 4.5

**Corollary 4.5.** *Following the assumptions and definitions in Corollary 4.4, for all $t \in [T]$,*

$$\mathbb{E}\|\text{TREE}(t)\|^2 \leq 2\ln(t)d\sigma^2 \leq \left(\frac{8\ln(T)d^{3/2}L}{B_2 T\rho}\right)^2.$$

*Proof.* Recall that $\text{TREE}(t) = \sum_{(\cdot, i)\in\text{NODE}(t)} \xi_i$. Since $\xi_i$'s are independent and $\xi_i \sim \mathcal{N}(0, \sigma_i^2 I)$,

$$\mathbb{E}\|\text{TREE}(t)\|^2 = \sum_{(\cdot, i)\in\text{NODE}(t)} \mathbb{E}\|\xi_i\|^2 = \sum_{(\cdot, i)\in\text{NODE}(t)} d\sigma_i^2 \leq 2\ln(t)d\sigma^2.$$

The last inequality follows from Remark 4.2 that $|\text{NODE}(t)| \leq 2\ln(t)$. We then conclude the proof by substituting the value of $\sigma$ defined in Corollary 4.4. $\qquad \square$

## D MISSING PROOFS IN SECTION 4.3

**Lemma 4.6.** *For any function $F : \mathbb{R}^d \to \mathbb{R}$ that is differentiable and $F(\boldsymbol{x}_1) - \inf_{\boldsymbol{x}} F(\boldsymbol{x}) \leq F^*$, if the domain of the OCO algorithm $\mathcal{A}$ is bounded by $D = \delta/T$, then*

$$\mathbb{E}\|\nabla F(\overline{\boldsymbol{w}})\|_\delta \leq \frac{F^*}{DKT} + \sum_{k=1}^K \frac{\mathbb{E}\text{Reg}_T(\boldsymbol{u}^k)}{DKT} + \sum_{k=1}^K \sum_{t=1}^T \frac{\mathbb{E}\langle \nabla F(\boldsymbol{w}_t^k) - \tilde{\boldsymbol{g}}_t^k, \Delta_t^k - \boldsymbol{u}^k\rangle}{DKT}.$$

*where $\boldsymbol{u}^k = -D\frac{\sum_{t=1}^T \nabla F(\boldsymbol{w}_t^k)}{\|\sum_{t=1}^T \nabla F(\boldsymbol{w}_t^k)\|}$.*

*Proof.* First, by fundamental theorem of calculus, we have

$$F(\boldsymbol{x}_{t+1}^k) - F(\boldsymbol{x}_t^k) = \int_0^1 \langle \nabla F(\boldsymbol{x}_t^k + t(\boldsymbol{x}_{t+1}^k - \boldsymbol{x}_t^k)), \boldsymbol{x}_{t+1}^k - \boldsymbol{x}_t^k\rangle \, dt$$

$$= \int_0^1 \langle \nabla F(\boldsymbol{x}_t^k + t\Delta_t^k), \Delta_t^k\rangle \, dt = \mathbb{E}_{s_t^k}\langle \nabla F(\boldsymbol{w}_t^k), \Delta_t^k\rangle.$$

Next, upon taking the telescopic sum (and recall $x_1^{k+1} = x_{T+1}^k$), we have

$$\mathbb{E}F(\boldsymbol{x}_{T+1}^K) - F(\boldsymbol{x}_1^1) = \sum_{k=1}^K \sum_{t=1}^T \mathbb{E}[F(\boldsymbol{x}_{t+1}^k) - F(\boldsymbol{x}_t^k)] = \sum_{k=1}^K \sum_{t=1}^T \mathbb{E}\langle \nabla F(\boldsymbol{w}_t^k), \Delta_t^k \rangle.$$

Note that $\langle \nabla F(\boldsymbol{w}_t^k), \Delta_t^k \rangle = \langle \nabla F(\boldsymbol{w}_t^k), \boldsymbol{u}^k \rangle + \langle \tilde{\boldsymbol{g}}_t^k, \Delta_t^k - \boldsymbol{u}^k \rangle + \langle \nabla F(\boldsymbol{w}_t^k) - \tilde{\boldsymbol{g}}_t^k, \Delta_t^k - \boldsymbol{u}^k \rangle$. Therefore, by definition of $\boldsymbol{u}^k$ and $\mathrm{Reg}_T(\boldsymbol{u}^k)$,

$$\sum_{t=1}^T \langle \nabla F(\boldsymbol{w}_t^k), \Delta_t^k \rangle = -D \left\| \sum_{t=1}^T \nabla F(\boldsymbol{w}_t^k) \right\| + \mathrm{Reg}_T(\boldsymbol{u}^k) + \sum_{t=1}^T \langle \nabla F(\boldsymbol{w}_t^k) - \tilde{\boldsymbol{g}}_t^k, \Delta_t^k - \boldsymbol{u}^k \rangle.$$

Upon rearranging the inequality and applying $F(\boldsymbol{x}_{T+1}^K) - F(\boldsymbol{x}_1^1) \leq F^*$, we have

$$\sum_{k=1}^K D\mathbb{E} \left\| \sum_{t=1}^T \nabla F(\boldsymbol{w}_t^k) \right\| \leq F^* + \sum_{k=1}^K \mathbb{E}\mathrm{Reg}_T(\boldsymbol{u}^k) + \mathbb{E}\sum_{k=1}^K \sum_{t=1}^T \langle \nabla F(\boldsymbol{w}_t^k) - \tilde{\boldsymbol{g}}_t^k, \Delta_t^k - \boldsymbol{u}_t^k \rangle.$$

Finally, note that $\|\boldsymbol{w}_t^k - \overline{w}^k\| \leq DT = \delta$, so $\|\nabla F(\overline{w}^k)\|_\delta \leq \|\frac{1}{T}\sum_{t=1}^T \nabla F(\boldsymbol{w}_t^k)\|$ by definition of $\|\cdot\|_\delta$. Moreover, $\mathbb{E}\|\nabla F(\overline{\boldsymbol{w}})\|_\delta = \mathbb{E}[\frac{1}{K}\sum_{i=1}^K \|\nabla F(\overline{w}^k)\|_\delta]$. Therefore, dividing both sides of the inequality by $DKT$ concludes the proof. $\qquad\square$

**Corollary 4.7.** *Suppose $F: \mathbb{R}^d \to \mathbb{R}$ is differentiable, $L$-Lipschitz, and $F(\boldsymbol{x}_1) - \inf_{\boldsymbol{x}} F(\boldsymbol{x}) \leq F^*$, and suppose the domain of $\mathcal{A}$ is bounded by $D = \delta/T$. Then*

$$\mathbb{E}\|\nabla F(\overline{\boldsymbol{w}})\|_{2\delta} \leq \frac{F^* + 2L\delta}{DKT} + \sum_{k=1}^K \frac{\mathbb{E}\mathrm{Reg}_T(\hat{\boldsymbol{u}}^k)}{DKT} + \sum_{k=1}^K \sum_{t=1}^T \frac{\mathbb{E}\langle \nabla \hat{F}_\delta(\boldsymbol{w}_t^k) - \tilde{\boldsymbol{g}}_t^k, \Delta_t^k - \hat{\boldsymbol{u}}^k \rangle}{DKT}.$$

*where $\hat{\boldsymbol{u}}^k = -D\frac{\sum_{t=1}^T \nabla \hat{F}_\delta(\boldsymbol{w}_t^k)}{\|\sum_{t=1}^T \nabla \hat{F}_\delta(\boldsymbol{w}_t^k)\|}$.*

*Proof.* By Lemma 2.2, $\hat{F}_\delta$ is differentiable. Next, since $\|\hat{F}_\delta - F\| \leq L\delta$,

$$\hat{F}_\delta(\boldsymbol{x}_1) - \inf_{\boldsymbol{x}} \hat{F}_\delta(\boldsymbol{x}) \leq F(\boldsymbol{x}_1) - \inf_{\boldsymbol{x}} F(\boldsymbol{x}) + 2L\delta = F^* + 2L\delta.$$

Hence, upon applying Lemma 4.6 on the uniform smoothing $\hat{F}_\delta$, we have

$$\mathbb{E}\|\hat{F}_\delta(\overline{\boldsymbol{w}})\|_\delta \leq \frac{F^* + 2L\delta}{DKT} + \sum_{k=1}^K \frac{\mathbb{E}\mathrm{Reg}_T(\hat{\boldsymbol{u}}^k)}{DKT} + \sum_{k=1}^K \sum_{t=1}^T \frac{\mathbb{E}\langle \nabla \hat{F}_\delta(\boldsymbol{w}_t^k) - \tilde{\boldsymbol{g}}_t^k, \Delta_t^k - \hat{\boldsymbol{u}}^k \rangle}{DKT}.$$

Finally, by Corollary 2.3, $\mathbb{E}\|F(\overline{\boldsymbol{w}})\|_{2\delta}$ is also bounded by RHS, which concludes the proof. $\qquad\square$

**Lemma 4.8.** *Suppose $f(\boldsymbol{x}, z)$ is differentiable and $L$-Lipschitz in $\boldsymbol{x}$ and the domain of $\mathcal{A}$ is bounded by $D = \delta/T$, then the variance of the gradient oracle $\mathcal{O}$ (Algorithm 4) is bounded by*

$$\mathbb{E}\|\nabla \hat{F}_\delta(\boldsymbol{w}_t^k) - \boldsymbol{g}_t^k\|^2 \leq \frac{16dL^2}{B_1} + \frac{64dL^2}{B_2 T}.$$

*Proof.* First, we recall a martingale concentration bound. Let $\boldsymbol{v}_1, \ldots, \boldsymbol{v}_n$ be a sequence of random vectors and let $\mathcal{F}_i = \sigma(\boldsymbol{v}_{1:i})$ be the $\sigma$-algebra generated by $\boldsymbol{v}_{1:i}$. If $\mathbb{E}[\boldsymbol{v}_i|\mathcal{F}_{i-1}] = 0$, then for all cross terms $i \leq j$, $v_i \in \mathcal{F}_{j-1}$ and thus $\mathbb{E}\langle \boldsymbol{v}_i, \boldsymbol{v}_j \rangle = \mathbb{E}_{\mathcal{F}_{j-1}}\langle \boldsymbol{v}_i, \mathbb{E}[\boldsymbol{v}_j|\mathcal{F}_{j-1}] \rangle = 0$. Consequently,

$$\mathbb{E}\|\textstyle\sum_{i=1}^n \boldsymbol{v}_i\|^2 = \textstyle\sum_{i=1}^n \mathbb{E}\|\boldsymbol{v}_i\|^2.$$

In our case, note that $\nabla \hat{F}(\boldsymbol{w}_t^k) = \nabla \hat{F}(\boldsymbol{w}_1^k) + \sum_{i=2}^t \nabla \hat{F}(\boldsymbol{w}_t^k) - \nabla \hat{F}(\boldsymbol{w}_{t-1}^k)$ and recall that $\boldsymbol{g}_t^k = \boldsymbol{g}_1^k + \sum_{i=2}^t \boldsymbol{d}_i$. Therefore, by the concentration inequality and Lemma 3.1 and 3.2,

$$\mathbb{E}\|\nabla \hat{F}(\boldsymbol{w}_t^k) - \boldsymbol{g}_t^k\|^2 \leq \mathbb{E}\|\nabla \hat{F}(\boldsymbol{w}_1^k) - \boldsymbol{g}_1^k\|^2 + \sum_{i=2}^t \mathbb{E}\|\nabla \hat{F}(\boldsymbol{w}_i^k) - \nabla \hat{F}(\boldsymbol{w}_{i-1}^k) - \boldsymbol{d}_i^k\|^2$$

$$\leq \frac{16dL^2}{B_1} + \sum_{i=2}^t \frac{16dL^2}{B_2 \delta^2} \mathbb{E}\|\boldsymbol{w}_i^k - \boldsymbol{w}_{i-1}^k\|^2.$$

We conclude the proof by Remark 4.1 that $\|\boldsymbol{w}_i^k - \boldsymbol{w}_{i-1}^k\| \leq 2D \leq \frac{2\delta}{T}$. $\qquad\square$

---

**Algorithm 6** Naive private gradient oracle

---

    **Input:** dataset $\mathcal{Z}$, constant $B$, noise $\sigma$
    **Initialize:** Partition $\mathcal{Z}$ into $KT$ subsets $Z_t^k$ of size $B$. The data size is $|\mathcal{Z}| = M = BKT$.
1:  Upon receiving round index $k, t$ and parameter $\boldsymbol{w}_t^k$:
2:  Sample $\boldsymbol{u}_1, \ldots, \boldsymbol{u}_B \sim \text{Uniform}(\mathbb{S})$ i.i.d.
3:  $\boldsymbol{g}_t^k \leftarrow \frac{1}{B} \sum_{i=1}^{B} \frac{d}{2\delta} (f(\boldsymbol{w}_t^k + \delta \boldsymbol{u}_i, z_i) - f(\boldsymbol{w}_t^k - \delta \boldsymbol{u}_i, z_i)) \boldsymbol{u}_i$ where $z_i$ is the $i$-th data in $Z_t^k$.
4:  Return $\tilde{\boldsymbol{g}}_t^k \leftarrow \boldsymbol{g}_t^k + \xi_t^k$ where $\xi_t^k \sim \mathcal{N}(0, \sigma^2 I)$ i.i.d.

---

# E   Comparison with naive approach

To concrete the discussion from Section 4.1, we'll delve into the analysis of the naive private algorithm built on O2NC. Specifically, this algorithm replaces the gradient oracle (as seen in line 5 of Algorithm 3) with a direct zeroth-order estimator, as defined in Algorithm 6. As a remark, $\boldsymbol{g}_t^k$ is an unbiased estimator and $\mathbb{E}\|\boldsymbol{g}_t^k\|^2 \lesssim L^2(\frac{d}{B} + 1)$, according to (Kornowski & Shamir, 2023) Remark 6. Moreover, following the same argument as the sensitivity bound in Lemma 3.1, the sensitivity of $\boldsymbol{g}_t^k$ is bounded by $O(dL/B)$. Consequently, if we set $\sigma = dL/B\rho$, each $\tilde{\boldsymbol{g}}_t^k$ achieves $(\alpha, \alpha\rho^2/2)$-RDP.

Although the algorithm appears more straightforward, its convergence analysis is less favorable. Following the proof idea of Theorem 4.9, we first recall the result in Corollary 4.7:

$$\mathbb{E}\|\nabla F(\overline{\boldsymbol{w}})\|_{2\delta} \leq \frac{F^* + 2L\delta}{DKT} + \sum_{k=1}^{K} \frac{\mathbb{E}\text{Reg}_T(\hat{\boldsymbol{u}}^k)}{DKT} + \sum_{k=1}^{K} \sum_{t=1}^{T} \frac{\mathbb{E}\langle \nabla \hat{F}_\delta(\boldsymbol{w}_t^k) - \tilde{\boldsymbol{g}}_t^k, \Delta_t^k - \hat{\boldsymbol{u}}^k \rangle}{DKT}.$$

Recall that $\mathbb{E}[\text{Reg}_T(\hat{\boldsymbol{u}}^k)] \leq D\sqrt{\sum_{t=1}^{T} \mathbb{E}\|\tilde{\boldsymbol{g}}_t^k\|^2}$. Therefore, we can bound the regret of OSD by

$$\mathbb{E}[\text{Reg}_T(\hat{\boldsymbol{u}}^k)] \leq D\sqrt{\sum_{t=1}^{T} \mathbb{E}\|\boldsymbol{g}_t^k\|^2 + \mathbb{E}\|\xi_t^k\|^2}$$
$$\lesssim D\sqrt{\sum_{t=1}^{T} L^2(\frac{d}{B} + 1) + \frac{d^3 L^2}{B^2 \rho^2}}$$
$$\lesssim DL\sqrt{T}\left(\frac{\sqrt{d}}{\sqrt{B}} + 1 + \frac{d^{3/2}}{B\rho}\right).$$

Next, we bound the sum of inner product. It's important to note that $\mathbb{E}\langle \nabla \hat{F}_\delta(\boldsymbol{w}_t^k) - \tilde{\boldsymbol{g}}_t^k, \Delta_t^k \rangle = 0$ because $\Delta_t^k$ is independent of $Z_t^k$. Therefore, using the martingale concentration bound, we have

$$\sum_{t=1}^{T} \mathbb{E}\langle \nabla \hat{F}_\delta(\boldsymbol{w}_t^k) - \tilde{\boldsymbol{g}}_t^k, \Delta_t^k - \hat{\boldsymbol{u}}^k \rangle = \mathbb{E}\langle \sum_{t=1}^{T} \nabla \hat{F}_\delta(\boldsymbol{w}_t^k) - \tilde{\boldsymbol{g}}_t^k, -\hat{\boldsymbol{u}}^k \rangle$$
$$\leq D\sqrt{\sum_{t=1}^{T} \mathbb{E}\|\nabla \hat{F}_\delta(\boldsymbol{w}_t^k) - \boldsymbol{g}_t^k\|^2 + \mathbb{E}\|\xi_t^k\|^2}$$
$$\lesssim DL\sqrt{T}\left(\frac{\sqrt{d}}{\sqrt{B}} + 1 + \frac{d^{3/2}}{B\rho}\right).$$

Upon substituting back into Corollary 4.7, we have

$$\mathbb{E}\|\nabla F(\overline{\boldsymbol{w}})\|_{2\delta} \lesssim \frac{F^* + L\delta}{DKT} + \frac{L}{\sqrt{T}}\left(\frac{\sqrt{d}}{\sqrt{B}} + 1 + \frac{d^{3/2}}{B\rho}\right)$$

Upon replacing $D = \delta/T$ and $M = KBT$, we have

$$\lesssim \frac{(F^* + L\delta)BT}{\delta M} + \frac{\sqrt{d}L}{\sqrt{BT}} + \frac{L}{\sqrt{T}} + \frac{d^{3/2}L}{B\sqrt{T}\rho}$$

To achieve the optimal non-private rate $O(d\delta^{-1}\epsilon^{-3})$, we need to set $B = 1$. Therefore, upon setting $T = \min\{(\frac{\sqrt{d}L\delta M}{F^* + L\delta})^{2/3}, (\frac{d^{3/2}L\delta M}{(F^* + L\delta)\rho})^{2/3}\}$, we have

$$\lesssim \left(\frac{(F^* + L\delta)dL^2}{\delta M}\right)^{1/3} + \left(\frac{(F^* + L\delta)d^{3/2}L^2}{\rho\delta M}\right)^{1/3}.$$

Equivalently, $\mathbb{E}\|\nabla F(\overline{\boldsymbol{w}})\|_{2\delta} \leq \epsilon$ if $M = \Omega\left(d(\frac{F^*L^2}{\delta\epsilon^3} + \frac{L^3}{\epsilon^3}) + d^{3/2}(\frac{F^*L^2}{\rho\delta\epsilon^3} + \frac{L^3}{\rho\epsilon^3})\right)$, whose dominating term is $\Omega(d^{3/2}\rho^{-1}\delta^{-1}\epsilon^{-3})$. In contrast, our carefully designed algorithm only requires $\tilde{\Omega}(d\delta^{-1}\epsilon^{-3} + d^{3/2}\rho^{-1}\delta^{-1}\epsilon^{-2})$ samples, as proved in Theorem 4.9. Notably, the naive algorithm suffers an additional cost of $\epsilon^{-1}$ for privacy.

