# OpenReview forum: "Private Zeroth-Order Nonsmooth Nonconvex Optimization"
_ICLR.cc/2024/Conference — ICLR 2024 poster_

### Official Review · Reviewer_pExn · 2023-10-13

**Soundness:** 4 excellent
**Presentation:** 4 excellent
**Contribution:** 3 good
**Rating:** 8
**Confidence:** 4

**Summary:**

The paper proposes a differentially-private zero-order algorithm for nonsmooth nonconvex optimization, and analyzes its convergence rate.

**Strengths:**

The paper is very well written overall.
The contribution fits naturally into the line of recent developments in nonsmooth nonconvex optimization, introducing the first private algorithm for this setting (as far as I know).
The techniques, though well rooted in previous works, are nontrivial to compose and are executed nicely.

**Weaknesses:**

The main weakness in my opinion is that most proof components (as far as I can tell, all but the trick of decomposing the gradient into a sum of differences and applying a grad-difference estimator) appear in previous works, adequately cited throughout the paper.

**Questions:**

Questions:
- Preliminaries, Definition of $\|\nabla h(x)\|_\delta$: Regarding the "equivalent" definition - why is this equivalent? This is not clear to me, unless repetitions in S are allowed. Can the authors please explain this? Only a bound in one direction is immediate, as far as I can tell (which suffices for the main result of the paper). Also, the inequality in the definition is probably meant to be an equality (though only the inequality is clear to me, as I previously mentioned).
- Lemma 2.2: I do not see why the "immediate corollary" is immediate. Can the authors please explain?

Minor comments:
- Section 1.1, 2nd paragraph typos: 1) "Out gradient oracle..."=Our; 2) Citing Dwork et al., Chan et al.: \citep is more suitale than \citet.
- Section 1.1, 3rd paragraph arrives too early in my opinion. The authors discuss technicalities which are completely unclear at this point of the paper.
- Preliminaries, Definition of $\|\nabla h(x)\|_\delta$: The infimum is of the norm over the set, currently miswritten as the set itself.
- Differential privacy preliminaries: The authors should define Renyi divergence, and ref the two stated facts - "then it is also...", and "ensures that \Acal is ...-RDP" (specifically cite the result, not the whole manuscript).
- More generally, regarding the differential privacy preliminaries paragraph: it is not clearly explained how differential privacy relates to the stochastic optimization model, this is important and should be revised. e.g., "two datasets Z,Z'\in\Zcal", what is \Zcal? It should be explained that this is the same z in the stochastic objective etc.
- Equation in online learning (standard OSD regret bound) - add citation.
- Lemma 2.2: Actually h does not need to be differentiable (it is automatically differentiable almost everywhere anyhow by Rademacher's theorem, which suffices for everything).
- Section 2.1, 1st paragraph: "... a low sensitivity gradient oracle from ...", maybe should emphasize that this is a *stochastic* gradient oracle.
- Section 3, paragraph second to last: "as Lin et al. has proved..." - As Lin et al. write before the proof of Lemma D.1, the lemma is actually taken from the paper "An optimal algorithm for bandit and zero-order convex optimization with two-point feedback" by Shamir [2017, Lemma 10], only restated by them. Hence this is a more appropriate reference.
- Paragraph before Remark 4.1 includes a forward reference to Remark 4.1, maybe the remark should appear earlier? (I do not see a reason not to.)
- Remark 4.2: I appreciate the authors adding an informal explanation about the Node function, though I found it hard to follow and I suggest revising it.
- Proof after Corollary 4.4 - this is a proof of Theorem 4.3, right? Should clarify this.

---

> ### Author Response · Authors · 2023-11-14
> **Rebuttal**
>
> We thank the reviewer for the careful reading and supportive rating. We also thank the reviewer for all the constructive comments, and we will address each comment accordingly in the future version. Below we address the reviewer’s questions.
>
> > **Q1.**
>
> As the reviewer pointed out, the definition of Goldstein stationary point implies the inequality $\\| \nabla h(x)\\|\_\delta \le \inf \\| \frac{1}{|S|}\sum_{y\in S} \nabla h(y)\\|$, but the converse is not necessarily true. In this sense, the word “Equivalently” should rather be “Consequently”. We apologize for the confusion caused by the mistake, and we will correct the wording.
>
> > **Q2.**
>
> First, we would like to quote Lin et. al. [1] Theorem 3.1: $\nabla \hat F_\delta(x) \in \partial_\delta F(x)$.
>
> By the definition that $\partial_\delta F(x) = \text{conv}(\cup_{y\in B(x,\delta)} \nabla F(y))$, any $g\in \partial_\delta \hat F_\delta(x)$ has form $g=\sum \lambda_i \hat \nabla F_\delta(y_i)$ where $\lambda_i$’s are convex coefficients and $y_i\in B(x,\delta)$. By the quoted Thm. 3.1, $\nabla \hat F_\delta(y_i) \in \partial_\delta F(y_i) \subseteq \partial_{2\delta} F(x)$ since $\\|x-y_i\\| \le \delta$. This further implies that $g \in \partial_{2\delta} F(x)$ for any $g\in \partial_\delta \hat F_\delta(x)$, and thus $\partial_\delta \hat F_\delta(x) \subseteq \partial_{2\delta} F(x)$. Consequently, $\\| \nabla F(x) \\|\_{2\delta} := \inf \\{ \\|g\\| : g\in \partial_{2\delta} F(x)\\} \le \inf \\{ \\|g\\| : g\in \partial_\delta \hat F_\delta(x)\\} := \\| \nabla \hat F_\delta(x) \\|_\delta$, and this proves Corollary 2.3.
>
> We realize this proof is not “immediately” from Lemma 2.2. We will address the wording for clarity and will include a formal proof of Cor. 2.3 in the future version.
>
> > **Last comment**
>
> We would like to clarify that the proof after Corollary 4.4 is the proof of Cor. 4.4 itself, while the proof of Theorem 4.3 is deferred to Appendix C due to space limit. We apologize for the confusion, and we will clarify this in the future version.

---

> > ### Comment · Reviewer_pExn · 2023-11-15
> > **Response to rebuttal**
> >
> > Thank you for the detailed and adequate response.

---

### Official Review · Reviewer_oGL7 · 2023-10-23

**Soundness:** 2 fair
**Presentation:** 3 good
**Contribution:** 2 fair
**Rating:** 5
**Confidence:** 4

**Summary:**

The paper studies differentially private zeroth-order algorithms for nonsmooth nonconvex optimization and proposes the first algorithm under this topic. The convergence analysis of the algorithm shows that the non-private term matches with the optimal rate in non-private zeroth-order nonsmooth nonconvex optimization. The private term is $d/\delta$ times worse than the best-known rate for private first-order algorithms for smooth nonconvex optimization.

**Strengths:**

The paper studies a new topic that aims to obtain private zeroth-order algorithms for nonsmooth nonconvex optimization, while most existing works for private nonconvex optimization focus on first-order algorithms for smooth objective functions. The proposed method creatively combines existing results and leads to a non-trivial convergence analysis. The presentation of the paper is well-structured and clearly introduces different components of this new algorithm.

**Weaknesses:**

1. The reason to study differentially private (DP) zeroth-order methods for nonsmooth nonconvex optimization is not well motivated in this paper. I agree it is important to study DP nonconvex optimization, and there is indeed a rich literature that focuses on first-order methods. The paper mentions some applications of zeroth-order methods where gradients can be hard to obtain, including reinforcement learning. However, there is no notion of the dataset in these applications, then it is not immediate to incorporate DP. The paper should find some settings where DP zeroth-order algorithms are necessary. The current introduction looks like a simple combination of different concepts.

2. The computation complexity is never reported and compared with existing DP first-order methods and non-DP zeroth-order methods. I guess it will be $d$ times worse since $d$ samples are required to construct the zeroth-order estimators, which could be bad when $d$ is large. I understand the need for $d$ samples to reduce the variance of DIFF, but why is it also used for GRAD? Also, the paper uses a two-point estimator for GRAD and a one-point estimator for each gradient in DIFF. Is there any specific reason for this difference in the choice?

3. Is the assumption that $f(x,z)$ is differentiable required? I think Lipschitzness is enough in most nonsmooth nonconvex optimization literature, which implies the function is almost everywhere differentiable. Moreover, the main convergence results rely on the assumption that the domain of $\mathcal{A}$ is bounded by $\delta/T$. Projection is thus required to make sure this assumption is satisfied. However, as $\delta/T$ is typically very small, such projection suggests that each update of $x$ only increases the magnitude of $x$ by at most $\delta/T$ and every iterate remains in a ball with radius $\delta$ centered at $x_1$ after $T$ updates. What if there is no stationary point in this ball? Does the convergence result still make sense? I guess it is then required that $\delta$ should be sufficiently large, and the $(\delta, \epsilon)$-stationarity would be weak.

4. I find it hard to parse Remark 4.2 and Theorem 4.3. What is the definition of $n_j$, $\mathcal{X}^i$ and $\mathcal{S}^{(i)}$? What is the need for the first index in the tuple of NODE? It is only required to sum the second index in NODE as per line 3 of Algorithm 5. I am also confused by the statement in Remark 4.2 that says NODE stores the largest node in each layer. NODE(7) and NODE(8) both have 3 layers, but one has 3 elements while the other only has 1 element.

5. Minor: The standard results in $(\epsilon, \delta)$-DP say something like $\Vert \nabla F\Vert^2 \leq \sqrt{d\log(1/\delta)}/(n\epsilon)$. Here, the dataset with size $n$ is given, and the best achievable rate given $n$ is studied. It might be good to also report such a rate following the standard since private data is not as many as one can collect and is assumed to be given in advance; The proposed method is a single-pass algorithm, but in machine learning practice, training tends to be in multiple epochs; In Section 2, $(\epsilon, \delta)$-DP is used when introducing differential privacy. It might be good to change it to avoid confusion with $(\delta, \epsilon)$-stationarity.

**Questions:**

See Weaknesses.

---

> ### Author Response · Authors · 2023-11-14
> **Rebuttal**
>
> We thank the reviewer for detailed reading and comments. We address the questions and comments below.
>
> > **Q3. discussion about $\delta$ and Online-to-nonconvex updates**
>
> We would like to clarify that our result **does not** have any implicit limitation on the range of $\delta$. Note that O2NC (Algorithm 3) has two loops: $K$ outer loops and $T$ inner loops for a total of $N=KT$ iterations. It is true that for each $T$ inner iterations, $x$ can move a distance up to $\delta$, i.e., $\\|x_1^k - x_{T+1}^k\\| \le \delta$. However, since there are $K$ such outer loops, the total distance between the last-iterate and the initial point is $\\|x_1^1 - x_{T+1}^K\\| \le K\delta$. As shown in Theorem 4.9, the optimal tuning of $K$ is approximately $N^{1/3}/d^{1/3}\delta^{2/3}$, which yields an upper bound of $K\delta = O((\delta N/d)^{1/3})$ instead of $O(\delta)$. We hope this clarifies the reviewer’s misunderstanding.
>
> > **Q2. computation complexity**
>
> The computation complexity of our algorithm is $O(dM)$ function evaluations, where $d$ is the dimension and $M$ is the data size. We admit that the computation complexity of our algorithm is $d$ times greater than first-order algorithms. However, we would like to point out that in practice, function evaluations can be much cheaper than gradient computations. Moreover, the task of computing $d$ i.i.d. function evaluations per data point can be further accelerated by parallelization.
>
> > **Q2. GRAD and DIFF estimators**
>
> We agree with the reviewer that it’s not necessary for the GRAD estimator to sample $d$ uniform vectors. Using the two-point estimator and following the previous result in (Shamir 2017) Lemma 10, we will achieve the same linear dimension dependence.
>
> For the DIFF estimator, our analysis technique does not require two-point estimators, as they would not enhance the dimension dependence of DIFF's variance. We provide a detailed discussion on the design of GRAD and DIFF in the last two paragraphs of Section 3.
>
> > **Q3. assumption that $f(x,z)$ is differentiable**
>
> We agree that assuming $f$ is differentiable is not required. For non-differentiable $f$, the online-to-non-convex algorithm (Cutkosky et. al. 2023) adds a small perturbation to $f$, achieving the same convergence guarantee. In our paper, we assumed differentiability to avoid this additional perturbation and thus simplify the presentation. Very likely in our setting even this perturbation is unnecessary (since we already perturb the loss).. We will add a discussion about differentiability in the future version.
>
> > **Q4. presentation of Remark 4.2 and Theorem 4.3**
>
> To clarify Remark 4.2 and Theorem 4.3, we would like to first reference our notation for binary trees detailed in Appendix C, which was deferred due to space limit. A node in a binary tree is denoted by the tuple $(i,j)$, where $i$ is the leftmost leaf child and $j$ is the rightmost leaf child. We agree that in the practical implementation of NODE,  the first index in the tuple is not required for summation. However, this tuple notation greatly simplifies the proof of Theorem 4.3.
>
> Furthermore, we would like to provide a more detailed explanation of the function $\text{NODE}(t)$. Consider a complete binary tree of $k=\lceil \log_2 t\rceil$ layers, with the root layer labeled as layer 0 and the leaf layer labeled as layer $k$. Starting from layer 0 which only contains the root node $(1,2^k)$, the root node is stored if $2^k\le t$, represented by the right index $n_0=2^k$. Otherwise, no node is stored in layer 0 and $n_0$ is set to $\varnothing$. In each subsequent layer $i$, a node with right index $n_i$ is stored if $n_i \le t$ and $n_i > n_j$ for all $j<i$. Specifically, **at most** one node is stored per layer, which explains why NODE(7) returns 3 nodes while NODE(8) only returns 1 node. The central message of Remark 4.2 is the inequality $|\text{NODE}(t)| \le \lceil \log_2 t\rceil$.
>
> For Theorem 4.3, $\mathcal{S}^{(i)}$ is a typo; it should instead be $\mathcal{X}$, representing the state space, i.e., the set of all parameters $x$ for $f(x,z)$ in our case. Here $\mathcal{X}^i$ denotes the Cartesian product of $\mathcal{X}$.
>
> We apologize for the confusion caused by our presentation, and we will improve the presentation with more clarity in the future version.
>
> **continues in the next thread...**

---

> ### Author Response · Authors · 2023-11-14
> **Rebuttal (part 2)**
>
> > **Q5. convergence bound in terms of data size $M$**
>
> In the proof of Theorem 4.9, we showed that $\mathbb{E} \\|\nabla F(\overline w)\\|_{2\delta} \lesssim (d/\delta M)^{1/3} + (d/M)^{1/3} + (d^{3/2}/\rho\delta M)^{1/2} + (d^{3/2}/\rho M)^{1/2}$. Here $\rho$ denotes the DP parameter (corresponding to $\epsilon$ in $(\epsilon,\delta)$-DP) and the symbol $\lesssim$ hides dependence on constants $F^*, L$ and logarithmic factors. In the future version, we will include this convergence bound w.r.t. $M$ in the main theorem.
>
> > **Q5. multi-epoch training**
>
> We want to clarify that our paper primarily addresses the stochastic optimization (SO) problem, which seeks to optimize the expected loss $\mathbb{E}[f(x,z)]$. In contrast, multi-epoch training pertains to the empirical risk minimization (ERM) problem, aiming to optimize the finite-sum loss $\frac{1}{n}\sum_{i=1}^n f(x,z_i)$. It is important to recognize that ERM, while of significant interest, represents a distinct topic. Moreover, prior works have indicated that SO and ERM problems exhibit differing convergence bounds [1].
>
>
> **References**
>
> [1] Arora, R., Bassily, R., González, T., Guzmán, C., Menart, M., and Ullah, E., “Faster Rates of Convergence to Stationary Points in Differentially Private Optimization”, *arXiv e-prints*, 2022. doi:10.48550/arXiv.2206.00846.

---

> > ### Comment · Reviewer_oGL7 · 2023-11-20
> >
> > My question 1 is not answered.
> >
> > I still have concerns regarding the projection step of $\mathcal{A}$ onto a ball of very small radius $\delta/T$. From a theoretical side, it is used to make sure $\Vert w_i - w_{i-1}\Vert^2$ can be bounded. However, previous standard analysis of variance reduced methods do not require such artificial projection step to control this difference. I wonder whether the same can be achieved here as well. From an implementation side, since the radius can be extremely small, some numerical problems might happen.
> >
> > My other questions are well addressed by the authors. Thanks a lot for your reply!

---

> > > ### Author Response · Authors · 2023-11-22
> > > **Response to Reviewer oGL7**
> > >
> > > We thank the reviewer for the discussion, and we answer the questions below.
> > >
> > > > **Q1. Motivation**
> > >
> > > Regarding motivation: In practice, zero-order optimization can be useful in cases where memory is constrained or the model is excessively large (in which case we do not have the resources to compute a backwards pass), or as mentioned before in tasks that have a RL aspect.
> > >
> > > In both cases it is reasonable to ask for privacy. In the first case (large models), the privacy motivation is the same as in any other standard supervised learning setting. In the RL case, one can imagine that at each step we provide the model to a set of different people who will report the model’s performance on whatever query they make, but not the gradients. Imagine, for example, a setting in which we present users with recommendations or chat responses, and get feedback in the form of “upvotes” or “downvotes”. We’d like to make updates without compromising the privacy of feedback from the individual users
> > >
> > > Beyond this, we feel the problem is inherently interesting from a mathematical point of view.
> > >
> > > > **Q2. Projection to a ball with radius $D = \delta/T$**
> > >
> > > To the best of our knowledge, no existing algorithm achieves the optimal convergence rate for non-smooth non-convex optimization without requiring the projection to a ball with radius $\delta/T$. For first-order algorithms, O2NC (Cutkosky et. al., 2023) achieves the optimal rate and requires this projection; for zeroth-order algorithms, the state-of-art result achieved by Kornowski & Shamir (2023) is built upon the O2NC framework and thus also requires the projection.
> > >
> > > For non-convex but smooth objectives, variance-reduction algorithms do not require such a projection. However, the non-smooth, non-convex optimization problem is significantly more challenging than its smooth counterpart. Consequently, the intuition in smooth optimization may not be directly transferable to this more complex problem. Devising an optimal algorithm that obviates the need for such projection remains an interesting open question even in the non-private setting.
> > >
> > > Furthermore, we would like to emphasize the fundamental role of this projection in finding a $(\delta,\epsilon)$-stationary point. To reiterate, a point $x$ is a $(\delta,\epsilon)$-stationary point if there exists $S\subset B(x,\delta)$ such that $\\|\frac{1}{|S|}\sum_{y\in S} \nabla F(y)\\| \le \epsilon$. In our framework, $x$ corresponds to $w^k = \frac{1}{T}\sum_{t=1}^T w_t^k$ and $S$ corresponds to $\{w_1^k,\ldots,w_T^k\}$. Therefore, ensuring that $\\|w_t^k - w_{t-1}^k\\| \le \delta/T$ is crucial so that $\\|w^k - w_t^k\\| \le \delta$ holds. Without the projection, the distance of $\\|w^k-w_t^k\\|$ cannot be bounded, thereby failing to satisfy the $(\delta,\epsilon)$-stationary point definition.

---

> > > > ### Comment · Reviewer_oGL7 · 2023-11-23
> > > >
> > > > Thanks for your detailed reply. Please include these discussions in the revised version. I increase my score to 5.

---

### Official Review · Reviewer_P8ng · 2023-11-03

**Soundness:** 3 good
**Presentation:** 4 excellent
**Contribution:** 3 good
**Rating:** 6
**Confidence:** 3

**Summary:**

This work introduces a zeroth-order stochastic optimization algorithm for nonconvex and nonsmooth objectives. This algorithm finds a $(\delta,\epsilon)$-stationary point with $(\alpha, \alpha \rho^2/2)$-Renyi differential privacy within $O(d {\delta^{-1} \epsilon^{-3}} + d^{3/2} \delta^{-1} {\rho^{-1} \epsilon^{-2}})$ data complexity.

This algorithm uses non-private Online-to-non-convex Conversion (O2NC) framework proposed in previous work that finds a $(\delta,\epsilon)$-stationary point using a first-order oracle. On top of this framework, this paper builds an approximate first-order oracle with a zeroth-order oracle. Specifically, this first-order oracle samples $d$ iid estimators for each data point to achieve optimal dependence on $d$.

**Strengths:**

1. This work investigates the important problem of nonconvex and nonsmooth optimization, which is a frequent setting in modern machine learning. It provides an efficient algorithm that finds a stationary point while attaining differential privacy. The sample complexity required matches its non-private analog.
2. The paper is technically solid, clearly presented, and well-structured, with the key proof step contained in the main text, so that it is easy for the readers to follow the key proof ideas.

**Weaknesses:**

1. As discussed in the paper, the need to sample $d$ iid estimators for each data point seems to be less natural.
2. It would be good to have some discussions on how differential privacy is attained in other similar (e.g., 1st order) optimization problems.

**Questions:**

1. It is claimed in Section 1.1 that "the non-private term $O(d {\delta^{-1} \epsilon^{-3}})$ matches the optimal rate found in its non-private counterpart." Is this sample complexity proved to be the lower bound for zeroth-order optimization on non-smooth non-convex objectives?
2. The algorithm samples $d$ iid estimators for each data point. Do you think it is essentially necessary to achieve the current dependence on $d$, or is it limited to the analysis technique?

---

> ### Author Response · Authors · 2023-11-14
> **Rebuttal**
>
> We thank the reviewer for their detailed reading and comments. We address the comments and questions below.
>
> > **Q2. sampling $d$ estimators per data point**
>
> The requirement of $d$ i.i.d. function evaluations per data point stems from the limit of our analysis technique in order to reduce the variance by a factor of $d$ and achieve the optimal dimension dependence. Exploring whether alternative methods can achieve the same dimension dependence without requiring $d$ function evaluations per data point, or showing the necessity of this requirement, remains an interesting open question. That said, we emphasize that these $d$ function evaluations are using the same sample, and so do not require more data samples.
>
> > **DP mechanisms in previous DP optimization algorithms**
>
> We thank the reviewer's constructive suggestion and will expand the discussion on DP mechanisms in the related works section. Here's a brief overview:
>
> DP guarantees in gradient-based optimization are typically achieved by adding Gaussian noise to stochastic gradients and then applying a non-DP algorithm. The noise variance depends on the sensitivity of these gradients, thus making the design of low-sensitivity gradient estimators a key focus in our paper. In Appendix E, we show that applying this approach to our setting results in significantly worse bounds. More advanced approaches in the literature use private amplification mechanisms to reduce noise levels while maintaining DP guarantees. These methods include amplification by subsampling [1], iteration [2], and shuffling [3], and are commonly applied in DP optimization algorithms to achieve better privacy guarantees. Particularly, the tree mechanism that we use is also one of those mechanisms for privately summing a sequence of updates [4].
>
> For illustration, DP-SGD [5] adds Gaussian noise to stochastic gradients of standard SGD and utilizes subsampling for amplification. Private SpiderBoost [6], building on the non-private SpiderBoost, employs the tree mechanism for noise reduction.
>
> > **Q1.**
>
> In Section 1.1, we intended to indicate that our rate matches the state-of-the-art, as demonstrated by Kornowski and Shamir (2023), and that the non-private component of the bound matches the optimal rate in terms of $\epsilon$ and $\delta$.. We'll revise our wording for clarity.
>
> Regarding the lower bound for zeroth-order non-smooth non-convex optimization, no formal proof exists for $\Omega(d\delta^{-1}\epsilon^{-3})$ to our knowledge. However, Cutkosky et. al. (2023) proved a $\delta^{-1}\epsilon^{-3}$ lower bound for first-order non-smooth non-convex problems, suggesting our algorithm is optimal in terms of $\delta,\epsilon$. Moreover, Duchi et. al. (2013) showed a $d\epsilon^{-2}$ lower bound for the zeroth-order convex and smooth problem with one function evaluation per iteration. This finding suggests that the lower bound of the zeroth-order non-smooth, non-convex problem is likely to exhibit some dimension dependence.
>
>
> **References**
>
> [1] Bassily, R., Smith, A., and Thakurta, A., “Differentially Private Empirical Risk Minimization: Efficient Algorithms and Tight Error Bounds”, *arXiv e-prints*, 2014. doi:10.48550/arXiv.1405.7085.
>
> [2] Feldman, V., Mironov, I., Talwar, K., and Thakurta, A., “Privacy Amplification by Iteration”, *arXiv e-prints*, 2018. doi:10.48550/arXiv.1808.06651.
>
> [3] Erlingsson, Ú., Feldman, V., Mironov, I., Raghunathan, A., Talwar, K., and Thakurta, A., “Amplification by Shuffling: From Local to Central Differential Privacy via Anonymity”, *arXiv e-prints*, 2018. doi:10.48550/arXiv.1811.12469.
>
> [4] Cynthia Dwork, Moni Naor, Toniann Pitassi, and Guy N. Rothblum. 2010. Differential privacy under continual observation. In Proceedings of the forty-second ACM symposium on Theory of computing (STOC '10). Association for Computing Machinery, New York, NY, USA, 715–724. https://doi.org/10.1145/1806689.1806787
>
> [5] Abadi, M., Chu, A., Goodfellow, I., McMahan, H. B., Mironov, I., Talwar, K., Zhang, L., “Deep Learning with Differential Privacy”, *arXiv e-prints*, 2016. doi:10.48550/arXiv.1607.00133.
>
> [6] Arora, R., Bassily, R., González, T., Guzmán, C., Menart, M., and Ullah, E., “Faster Rates of Convergence to Stationary Points in Differentially Private Optimization”, *arXiv e-prints*, 2022. doi:10.48550/arXiv.2206.00846.
>
> [7] Kornowski, G. and Shamir, O., “An Algorithm with Optimal Dimension-Dependence for Zero-Order Nonsmooth Nonconvex Stochastic Optimization”, *arXiv e-prints*, 2023. doi:10.48550/arXiv.2307.04504.
>
> [8] Cutkosky, A., Mehta, H., and Orabona, F., “Optimal Stochastic Non-smooth Non-convex Optimization through Online-to-Non-convex Conversion”, *arXiv e-prints*, 2023. doi:10.48550/arXiv.2302.03775.
>
> [9] Duchi, J. C., Jordan, M. I., Wainwright, M. J., and Wibisono, A., “Optimal rates for zero-order convex optimization: the power of two function evaluations”, *arXiv e-prints*, 2013. doi:10.48550/arXiv.1312.2139.

---

> > ### Comment · Reviewer_P8ng · 2023-11-22
> >
> > Thanks for the detailed responses. My questions are addressed, and I keep the score for this work.

---

### Official Review · Reviewer_oD4K · 2023-11-05

**Soundness:** 3 good
**Presentation:** 3 good
**Contribution:** 3 good
**Rating:** 6
**Confidence:** 4

**Summary:**

This paper studies the problem of zeroth-order nonsmooth nonconvex optimization with differential privacy and provides an algorithm with sample complexity $\tilde{O}(\frac{d}{\delta\epsilon^3} + \frac{d^{3/2}}{\rho\delta\epsilon^2})$, where the optimal sample complexity for the non-private version is $\tilde{O}(\frac{d}{\delta\epsilon^3})$. The authors obtain this result by constructing a variance-reduced oracle with the tree mechanism.

**Strengths:**

The paper is technically solid. The private setting is both important and interesting. This was done without sacrificing a good complexity; the sample complexity is optimal in some regimes $\rho > \sqrt{d} \epsilon$. In general, I think this is a nice paper with a clear idea and with good explanations that show the motivation for each statement or its proof. I really like the authors explained why the naive approach would give the sub-optimal rate in the appendix.

**Weaknesses:**

1. The result heavily based on the previous paper on the non-private version (Cutkosky et al., 2023.) makes it seem a bit incremental.
2. Understandably, there is no space in the current ICLR format for experimental evaluation, this is something that could be looked at in the future.

**Questions:**

-

---

> ### Author Response · Authors · 2023-11-14
> **Rebuttal**
>
> We thank the reviewer for their detailed reading and comments. We address the comments below, and we are glad to answer any future questions.
>
> We agree with the reviewer that our algorithm builds on the non-private online-to-non-convex (O2NC) conversion. However, adapting O2NC for privacy involves several complex techniques, such as creating low-sensitivity zeroth-order estimators and using the tree mechanism to reduce noise. Even these need some finesse to be used in our setting - as evidenced by our use of multiple function evaluations to reduce variance in the gradient difference estimator. The technical novelty in our work lies in developing the insights needed to combine these techniques to achieve a private algorithm. Moreover, we would also like to emphasize that our research is the first to solve the zeroth-order non-smooth, non-convex optimization problem with differential privacy.

---

### Meta-Review · Area_Chair_roKy · 2023-12-06

**Metareview:**

The paper introduces a zeroth-order stochastic optimization algorithm for nonconvex and nonsmooth objectives. The reviewers think that the paper is technically solid and the private setting is both important and interesting. The authors have addressed well the reviewers' concerns. Therefore, most of the reviewers support the publication of the paper.

**Justification For Why Not Higher Score:**

The reviewers do not strongly support the paper.

**Justification For Why Not Lower Score:**

Most of the reviewers support the paper.

---

### Decision · Program_Chairs · 2024-01-16

Accept (poster)